# Human mitochondrial carriers of the SLC25 family function as monomers exchanging substrates with a ping-pong kinetic mechanism

Camila Cimadamore-Werthein[1,2], Martin S King [1,2], Denis Lacabanne [1], Eva Pyrihová[1], Stephany Jaiquel Baron[1] & Edmund RS Kunji [1✉]

## Abstract

Members of the SLC25 mitochondrial carrier family link cytosolic and mitochondrial metabolism and support cellular maintenance and growth by transporting compounds across the mitochondrial inner membrane. Their monomeric or dimeric state and kinetic mechanism have been a matter of long-standing debate. It is believed by some that they exist as homodimers and transport substrates with a sequential kinetic mechanism, forming a ternary complex where both exchanged substrates are bound simultaneously. Some studies, in contrast, have provided evidence indicating that the mitochondrial ADP/ATP carrier (SLC25A4) functions as a monomer, has a single substrate binding site, and operates with a ping-pong kinetic mechanism, whereby ADP is imported before ATP is exported. Here we reanalyze the oligomeric state and kinetic properties of the human mitochondrial citrate carrier (SLC25A1), dicarboxylate carrier (SLC25A10), oxoglutarate carrier (SLC25A11), and aspartate/glutamate carrier (SLC25A13), all previously reported to be dimers with a sequential kinetic mechanism. We demonstrate that they are monomers, except for dimeric SLC25A13, and operate with a ping-pong kinetic mechanism in which the substrate import and export steps occur consecutively. These observations are consistent with a common transport mechanism, based on a functional monomer, in which a single central substrate-binding site is alternately accessible.

**Keywords** SLC25 Mitochondrial Carrier Family; Kinetic Analysis; Bioenergetics; Mitochondria; Transport
**Subject Categories** Membranes & Trafficking; Organelles

## Introduction

The SLC25 mitochondrial carrier family has 53 members, forming the largest solute transporter family in humans (Kunji et al, 2020). Its members play crucial roles in cellular physiology by transporting a diverse set of substrates across the impermeable mitochondrial inner membrane, connecting cytosolic and mitochondrial metabolic pathways, and supporting cellular and mitochondrial processes for maintenance and growth (Kunji et al, 2020). Even though they were discovered nearly sixty years ago (Chappell, 1968; Pfaff et al, 1965), the fundamental mechanism by which they operate is still being investigated (Kunji et al, 2020; Ruprecht and Kunji, 2020). The most detailed information about the structural mechanism is available for the mitochondrial ADP/ATP carrier, also called adenine nucleotide translocase (Ruprecht and Kunji, 2019, 2021). The mitochondrial ADP/ATP carrier is a structural and functional monomer (Bamber et al, 2006; Bamber et al, 2007a; Bamber et al, 2007b; Kunji and Harding, 2003; Kunji et al, 2008; Mifsud et al, 2013; Pebay-Peyroula et al, 2003; Ruprecht et al, 2014; Ruprecht et al, 2019). The carrier has a single central substrate-binding site (Kunji and Robinson, 2006; Mavridou et al, 2022; Robinson and Kunji, 2006; Robinson et al, 2008; Ruprecht et al, 2019) and two gates with salt bridge networks and braces regulating access to the binding site from both sides of the membrane (King et al, 2016; Robinson et al, 2008; Ruprecht et al, 2014; Ruprecht et al, 2019; Springett et al, 2017). Recently, we have shown that the human mitochondrial ADP/ATP carrier (SLC25A4) operates with a ping-pong or double-displacement kinetic mechanism in which ADP is imported before ATP is exported (Cimadamore-Werthein et al, 2023). Thus, the kinetic and structural data are consistent with an alternating-access mechanism, based on the carrier functioning as a monomer (Bamber et al, 2007a; Cimadamore-Werthein et al, 2023).

However, this transport mechanism is fundamentally different from those reported for other members of the SLC25 mitochondrial carrier family, which are claimed to be homodimers, operating with a sequential kinetic mechanism (Palmieri et al, 1993). In a sequential mechanism, a ternary complex is formed where both exchanged substrates bind to the carrier simultaneously before transport in opposite directions occurs. These claims have been consistently made for several mitochondrial carriers with different substrate specificities. The mitochondrial citrate carrier (CIC, SLC25A1) catalyzes the electroneutral exchange of citrate or isocitrate for a tricarboxylate, a dicarboxylate, or phosphoenolpyruvate (Majd et al, 2017; Robinson et al, 1971). It has been proposed

[1]Medical Research Council Mitochondrial Biology Unit, The Keith Peters Building, Cambridge Biomedical Campus, Hills Road, Cambridge CB2 0XY, United Kingdom. [2]These authors contributed equally: Camila Cimadamore-Werthein, Martin S King. ✉E-mail: ek@mrc-mbu.cam.ac.uk

that CIC operates by a sequential kinetic mechanism (Bisaccia et al, 1993; Bisaccia et al, 1990) and is homodimeric based on blue native gel electrophoresis (Capobianco et al, 2002) and size exclusion chromatography (Kotaria et al, 1999). The mitochondrial dicarboxylate carrier (DIC, SLC25A10) transports dicarboxylates, such as malate, malonate, and succinate, as well as inorganic ions, such as phosphate, sulfate, sulfite, and thiosulfate (Chappell and Haarhoff, 1967; Fiermonte et al, 1999; Johnson and Chappell, 1973; Pyrihová et al, 2024; Szewczyk et al, 1987). DIC was proposed to have a sequential kinetic mechanism (Indiveri et al, 1989a; Indiveri et al, 1989b; Indiveri et al, 1993) and a dimeric state based on blue native gel electrophoresis (Palmieri et al, 1999). The mitochondrial oxoglutarate carrier (OGC, SLC25A11) transports 2-oxoglutarate into the mitochondrial matrix in exchange for malate (Bisaccia et al, 1985; Iacobazzi et al, 1992; Robinson and Chappell, 1967). Other substrates of this carrier include oxaloacetate, malonate, 2-oxoadipate, and malate (Pyrihová et al, 2024). Reconstituted OGC was suggested to have a sequential kinetic mechanism (Indiveri et al, 1991) in agreement with data obtained using intact mitochondria (Sluse et al, 1973). OGC has also been claimed to be a homodimer based on blue native gel electrophoresis (Palmisano et al, 1998) and cross-linking experiments (Bisaccia et al, 1996). Finally, the mitochondrial aspartate/glutamate carrier (AGC1, SLC25A12 or AGC2, SLC25A13) imports glutamate together with a proton into the mitochondrial matrix and exports aspartate to the intermembrane space (Azzi et al, 1967; Dierks and Kramer, 1988; Palmieri et al, 2001). AGC has a tripartite structure formed by a calcium-bound N-terminal domain, a mitochondrial carrier domain, and a C-terminal domain that interacts with the N-terminal domain (Thangaratnarajah et al, 2014). AGC is a structural homodimer through interactions between the N-terminal domains rather than the carrier domains, as shown by structural analyses and size exclusion chromatography combined with multi-angle laser light scattering (Thangaratnarajah et al, 2014). When the effect of internal glutamate on aspartate transport was tested in submitochondrial particles, a ping-pong kinetic mechanism was proposed (LaNoue et al, 1979), but reconstituted AGC was reported to have a sequential mechanism (Dierks et al, 1988).

Since mitochondrial carriers have the same structural and functional elements (Kunji et al, 2020; Ruprecht and Kunji, 2020), it seems unlikely that they would have fundamentally different oligomeric states and kinetic mechanisms. Here, we reanalyze the oligomeric state and kinetic properties of the aforementioned human carriers: OGC, CIC, DIC, and AGC2. For this purpose, the carriers were expressed in the mitochondrial inner membrane of *Saccharomyces cerevisiae* (King and Kunji, 2020) and purified in lauryl maltose neopentyl glycol, supplemented with cardiolipin to provide optimal stabilizing conditions (Crichton et al, 2015). Their oligomeric states were reexamined by size exclusion chromatography, showing that they are all monomeric except for AGC2, which is confirmed here to be a dimer. To study their kinetic properties, the carriers were reconstituted into liposomes and robotic transport assays were performed, where transport curves were recorded for different imposed substrate concentration gradients. Our data support the conclusion that all of these mitochondrial carriers, like the mitochondrial ADP/ATP carrier (Cimadamore-Werthein et al, 2023), operate with a ping-pong kinetic mechanism, in contrast to earlier claims.

# Results

## Expression, purification, stabilization, and reconstitution of human mitochondrial carriers

To study the kinetic mechanism of the human mitochondrial oxoglutarate carrier (OGC), citrate carrier (CIC), dicarboxylate carrier (DIC), and aspartate/glutamate carrier 2 (AGC2), we have adapted the protocols previously used for the mitochondrial ADP/ATP carrier (Cimadamore-Werthein et al, 2023). We expressed OGC (Fig. 1A), CIC (Fig. 1E), DIC (Fig. 1I), and AGC2 (Fig. 1M) in the mitochondrial inner membrane of *Saccharomyces cerevisiae* (King and Kunji, 2020; Pyrihová et al, 2024; Thangaratnarajah et al, 2014), and purified them in the detergent lauryl maltose neopentyl glycol (Chae et al, 2010), supplemented with cardiolipin to enhance their stabilities (Crichton et al, 2015). We first used thermostability assays to assess whether the purified proteins were folded and able to bind substrate. In nano differential scanning fluorimetry, the spectral properties of tryptophan and tyrosine residues shift as their local environment changes during heat denaturation (Alexander et al, 2014). The assay produces an apparent melting temperature, which corresponds to the temperature at which approximately half of the protein population has unfolded (Majd et al, 2018; Mavridou et al, 2022). The apparent melting temperatures are 51.0 °C for OGC (Fig. 1B), 54.3 °C for CIC (Fig. 1F), 54.1 °C for DIC (Fig. 1J), and 55.4 °C for AGC2 (Fig. 1N), similar to those observed for other mitochondrial carrier proteins (Crichton et al, 2015; Jaiquel Baron et al, 2021; Majd et al, 2018; Pyrihová et al, 2024). We have previously shown that both inhibitors (Crichton et al, 2015; Tavoulari et al, 2022) and substrates (Majd et al, 2018; Mavridou et al, 2022; Pyrihová et al, 2024) can specifically enhance the stability of a protein population in thermal denaturation assays, causing a thermostability shift. Consistent with these observations, substrate-dependent thermal shifts were observed for all proteins: 5.6 °C for malate binding to OGC (Fig. 1B), 7.8 °C for citrate binding to CIC (Fig. 1F), 2.5 °C for malate binding to DIC (Fig. 1J), and 1.6 °C for aspartate binding to AGC2 (Fig. 1N). These results show that the purified proteins are folded and interact with their known substrates in solution. We also tested whether the proteins are capable of transport. For this purpose, unlabeled substrate was incorporated into proteoliposomes by freeze-thaw-extrusion (Jaiquel Baron et al, 2021; Pyrihová et al, 2024), and external, non-internalized substrate was removed by gel filtration. Transport of substrates was initiated by the addition of external radiolabeled substrate and uptake curves were recorded by monitoring the accumulation of radiolabel inside proteoliposomes. At 1 mM internal and 2.5 μM external substrate, initial transport rates of 190 nmol [$^{14}$C]-malate mg$^{-1}$ min$^{-1}$ for OGC (Fig. 1C,D), 24 nmol [$^{14}$C]-citrate mg$^{-1}$ min$^{-1}$ for CIC (Fig. 1G,H), 30 nmol [$^{14}$C]-malate mg$^{-1}$ min$^{-1}$ for DIC (Fig. 1K,L), and 52 nmol [$^{14}$C]-aspartate mg$^{-1}$ min$^{-1}$ for AGC (Fig. 1O,P) were measured, demonstrating that all four reconstituted proteins were active.

## Mitochondrial carriers are monomeric except for the aspartate/glutamate carrier

Models of the carriers, determined by Alphafold 3.0 (Abramson et al, 2024) or experimentally (Jones et al, 2023), show the typical SLC25 carrier fold (Fig. 2A) (Kunji et al, 2020; Ruprecht and Kunji,

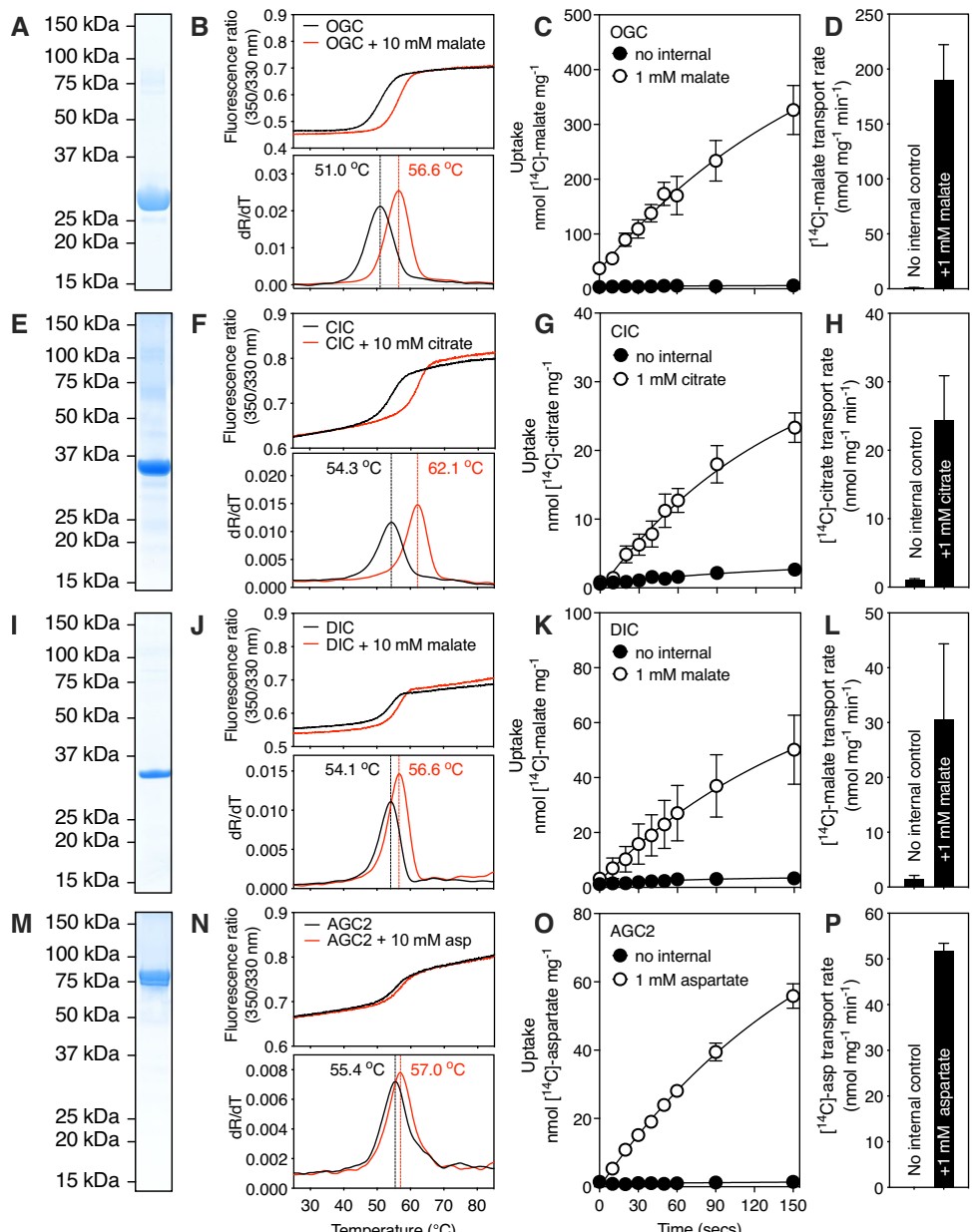

**Figure 1. The human oxoglutarate, citrate, dicarboxylate, and aspartate/glutamate carriers are pure, folded, bind substrate, and are functional.**

Instant-blue stained SDS-PAGE gel of purified protein (**A**) oxoglutarate carrier (OGC), (**E**) citrate carrier (CIC), (**I**) dicarboxylate carrier (DIC), (**M**) aspartate/glutamate carrier (AGC2). Typical unfolding curves of ~10 μg protein without compound (black trace), or with 10 mM compound (red trace) (**B**) OGC + 10 mM malate, (**F**) CIC + 10 mM citrate, (**J**) DIC + 10 mM malate, or (**N**) AGC2 + 10 mM aspartate, using nano differential scanning fluorimetry (Nanotemper Prometheus). The peak in the derivative of the unfolding curve (dR/dT) is at the apparent melting temperature (Tm). (**C**) [$^{14}$C]-malate uptake curves of OGC reconstituted into proteoliposomes loaded with (white circles) or without (black circles) 1 mM malate. Transport was initiated by the external addition of 2.5 μM [$^{14}$C]-malate. (**G**) [$^{14}$C]-citrate uptake curves of CIC reconstituted into proteoliposomes loaded with (white circles) or without (black circles) 1 mM citrate. Transport was initiated by the external addition of 2.5 μM [$^{14}$C]-citrate. (**K**) [$^{14}$C]-malate uptake curves of DIC reconstituted into proteoliposomes loaded with (white circles) or without (black circles) 1 mM malate. Transport was initiated by the external addition of 2.5 μM [$^{14}$C]-malate. (**O**) [$^{14}$C]-aspartate uptake curves of AGC2 reconstituted into proteoliposomes loaded with (white circles) or without (black circles) 1 mM aspartate. Transport was initiated by the external addition of 2.5 μM [$^{14}$C]-aspartate. (**D, H, L, P**) Initial rates were estimated by fitting the uptake curves to Eq. 1. The data represent the average and standard deviation of two biological repeats, each with three technical repeats except AGC2, which is the average of three technical repeats. Source data are available online for this figure.

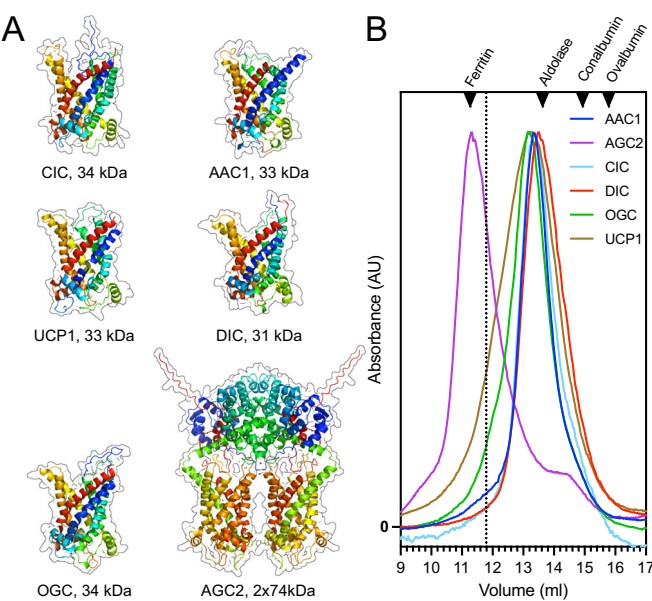

**Figure 2. Structures and oligomeric states of mitochondrial carriers.**

(A) Structural models of the human dicarboxylate carrier (DIC), oxoglutarate carrier (OGC), citrate carrier (CIC), ADP/ATP carrier (AAC1), and aspartate/glutamate carrier (AGC2) were all determined by the Alphafold 3.0 server (Abramson et al, 2024), except for the experimentally determined structure of an uncoupling protein (UCP1) (pdb entry: 8G8W) (Jones et al, 2023). (B) Determination of the molecular weights in LMNG/TOCL by size exclusion chromatography. The normalized absorbance traces for monomeric AAC1 (blue), DIC (red), OGC (green), CIC (cyan), UCP1 (brown), and dimeric AGC2 (purple) are shown. The standards used for sizing were ferritin (440 kDa), aldolase (158 kDa), conalbumin (76 kDa), and ovalbumin (43 kDa). The dotted line indicates the elution volume of a hypothetical dimer peak for DIC, OGC, and CIC. Source data are available online for this figure.

2020, 2021). AGC2 is unique, as it has an additional N-terminal domain involved in the dimerization of the protomers (Thangaratnarajah et al, 2014). To obtain experimental data on the oligomeric state, we analyzed the molecular weight of the purified proteins in detergent by size exclusion chromatography (Fig. 2B), using the stabilizing conditions in which the proteins were shown to be folded (Fig. 1B,F,J,N). The results show that OGC, CIC, and DIC elute in the same volume as the mitochondrial ADP/ATP carrier (Bamber et al, 2006; Bamber et al, 2007b; Kunji and Harding, 2003) and uncoupling protein (Jones et al, 2023; Lee et al, 2015), both of which have been demonstrated to be monomeric in detergent. When the total weight is corrected for the contribution of bound lipid/detergent (≈135 KDa), as done previously (Bamber et al, 2006; Kunji et al, 2008), the determined molecular weights (~30 kDa) corresponded well with those based on the amino acid sequences (31–34 KDa). In contrast, AGC2 elutes earlier and is confirmed here to be a dimer (~180 kDa), which occurs through interactions of the N-terminal domains of the two protomers (74 kDa each) (Thangaratnarajah et al, 2014). These analyses also show that OGC, CIC, and DIC are not naturally cross-linked when isolated from the mitochondrial inner membrane. Thus, in contrast to earlier claims, all tested mitochondrial carriers are monomeric,

except for AGC2, which is confirmed to be a dimer (Thangaratnarajah et al, 2014).

## Mitochondrial carriers transport with a ping-pong kinetic mechanism

Next, we studied the kinetic mechanism of this set of mitochondrial carriers by carrying out bi-reactant initial-velocity studies (Cleland, 1963a, b, 1973). In a sequential mechanism, the $K_m/V_{max}$ ratio decreases as the concentration of the counter-substrate increases, resulting in converging Lineweaver–Burk plots, whereas in a ping-pong or double-displacement kinetic mechanism, the $K_m/V_{max}$ ratio is independent of the concentration of counter-substrate, resulting in parallel Lineweaver–Burk plots.

Mitochondrial carriers are fully reversible transporters and can catalyze hetero-exchange in which the exchanged substrates are chemically different and homo-exchange in which they are chemically the same. To avoid unnecessary complications, caused by unequal charges or proton coupling, we carried out homo-exchange experiments. To determine the initial uptake rates for a range of different concentration gradients of substrates, we loaded proteoliposomes with either no substrate, which served as a background control, or with four different internal substrate concentrations. After removal of the external unlabeled substrate, we initiated the exchange with seven (AGC2) or eight (OGC, CIC, DIC) different concentrations of external radiolabeled substrate (Fig. 3A,D,G,J). The initial rates for each of the substrate gradients were obtained by fitting the uptake curves (Figs. EV1–EV4) with the following equation:

$$Q_i = \beta \left( 1 - e^{-\frac{k}{\beta}(t-d)} \right) \qquad (1)$$

Where $Q_i$ is the quantity of radiolabeled substrate on the inside of the proteoliposomes, $\beta$ is the total quantity of substrate exchanged at equilibrium, $k$ is the initial influx rate of radiolabeled substrate, $t$ is the exchange period, and $d$ is the time delay due to filtration (Cimadamore-Werthein et al, 2023).

By fitting the data to Eq. 1, accurate estimates of the initial uptake rates, given by $k$, can be obtained. These initial rates were then used to construct Michaelis–Menten curves and Lineweaver–Burk plots for each experiment (Fig. 3). The Lineweaver–Burk plots for OGC (Fig. 3B), CIC (Fig. 3E), DIC (Fig. 3H), and AGC2 (Fig. 3K) all appear to be parallel by visual inspection, which is consistent with a ping-pong rather than a sequential kinetic mechanism. However, Lineweaver–Burk plot representations in kinetic analyses are non-optimal due to unequal error distribution. A better way to analyze the kinetic mechanism is to obtain the apparent $K_m$ and $V_{max}$ values independently by fitting the Michaelis–Menten curves by iteration and by plotting the determined kinetic parameters against each other (Cimadamore-Werthein et al, 2023). Analyzed in this way, the data show that the $K_m/V_{max}$ ratio for all measured internal concentrations is constant within experimental error for all tested carriers: OGC (Fig. 3C), CIC (Fig. 3F), DIC (Fig. 3I), and AGC2 (Fig. 3L). These results demonstrate conclusively that these human mitochondrial carriers operate with a ping-pong or double-displacement kinetic mechanism.

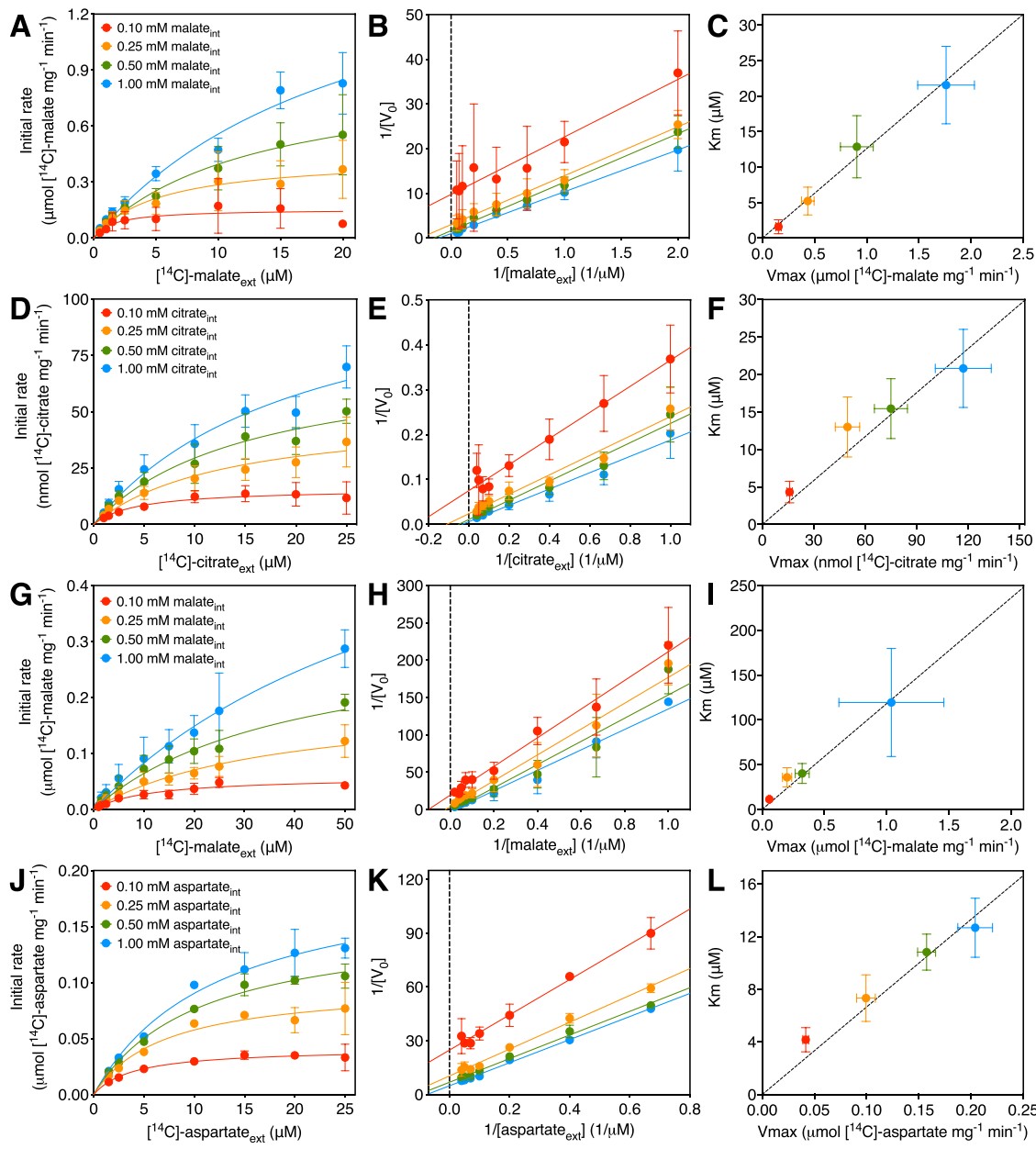

**Figure 3. Two-substrate analysis of transport catalyzed by human mitochondrial carriers.**

Initial rates were estimated by fitting uptake data to Eq. 1: OGC (Figs. 1C and EV1), CIC (Figs. 1G and EV2), DIC (Figs. 1K and EV3), and AGC2 (Figs. 1O and EV4). Michaelis–Menten plots of 0.10 mM (red traces), 0.25 mM (orange traces), 0.50 mM (green traces), and 1.00 mM (blue traces) internally-loaded (**A**) malate (OGC), (**D**) citrate (CIC), (**G**) malate (DIC), and (**J**) aspartate (AGC2). Transport was initiated by the addition of radiolabeled substrate externally ([$^{14}$C]-malate (OGC and DIC), [$^{14}$C]-citrate (CIC), and [$^{14}$C]-aspartate (AGC2)) at the indicated concentration, and stopped by vacuuming and washing at defined time-intervals. Lineweaver–Burk analysis of homo-exchange for (**B**) OGC, (**E**) CIC, (**H**) DIC, and (**K**) AGC2, using the same color scheme as in (**A**). $K_m$ plotted against $V_{max}$ for the various chemical gradients for (**C**) OGC, (**F**) CIC, (**I**) DIC, and (**L**) AGC2, using the same color scheme as in (**A**). The two kinetic parameters were determined by fitting the Michaelis–Menten curves through iteration. The data were represented by the average and standard deviation of $n = 6$ (two independent experiments, each with three technical repeats, except for the 1 and 50 μM external [$^{14}$C]-malate datasets for DIC, the 1.5 μM external [$^{14}$C]-malate/1 mM internal malate datapoint for DIC, and the data for AGC2, which are the average of three technical repeats). Source data are available online for this figure.

## Discussion

Before the structures were known, it was widely believed that mitochondrial carriers were homodimers operating with a sequential mechanism in which the two exchanged substrates are bound to the transporting unit simultaneously as part of the transport cycle (Palmieri et al, 1993). This kinetic mechanism had been proposed for the mitochondrial oxoglutarate carrier (Indiveri et al, 1991), citrate carrier (Bisaccia et al, 1993), dicarboxylate carrier (Indiveri et al, 1993), and aspartate/glutamate carrier (Dierks et al, 1988).

There were conflicting claims with regard to the mitochondrial aspartate/glutamate carrier, for which a ping-pong mechanism was obtained in whole mitochondria (LaNoue et al, 1979) versus a sequential mechanism in proteoliposomes (Dierks et al, 1988). There were also claims that these mitochondrial carriers are all dimeric (Bisaccia et al, 1996; Capobianco et al, 2002; Kotaria et al, 1999; Palmieri et al, 1999; Palmisano et al, 1998). By combining the homodimer and kinetic models, a structural mechanism was proposed in which the two protomers of the dimer each bind a substrate before they are exchanged across the membrane (Palmieri et al, 1993).

However, this transport model was inconsistent with the first structural information of the mitochondrial ADP/ATP carrier in the membrane, showing that the carrier is a structural monomer with threefold pseudo-symmetry (Kunji and Harding, 2003), reflecting the three homologous sequence repeats (Saraste and Walker, 1982). Since the substrate translocation pathway was through the center of the protein, it suggested for the first time that the carrier could be functional as a monomer (Kunji and Harding, 2003). Subsequent atomic-resolution structures confirmed that the structural fold of mitochondrial carriers is indeed monomeric (Jones et al, 2023; Pebay-Peyroula et al, 2003; Ruprecht et al, 2014; Ruprecht et al, 2019). A careful reanalysis, which had the benefit of hindsight, identified several technical issues that led to wrong interpretations in favor of a dimer, when in reality, a monomer was observed in all cases (Kunji and Crichton, 2010; Kunji and Ruprecht, 2020). Here we demonstrate that the mitochondrial oxoglutarate, dicarboxylate, and citrate carriers elute in size exclusion chromatography in the same fraction as the mitochondrial ADP/ATP carrier (Bamber et al, 2006; Bamber et al, 2007b; Kunji and Harding, 2003; Ruprecht et al, 2014) and uncoupling protein (Jones et al, 2023; Lee et al, 2015), both of which have been shown to be structural and functional monomers. The only confirmed exception is the mitochondrial aspartate/glutamate carrier, which dimerizes through the interaction of its additional N-terminal domain (Thangaratnarajah et al, 2014). The previous incorrect assignment of the oligomeric state in blue native gel electrophoresis (Capobianco et al, 2002; Palmieri et al, 1999; Palmisano et al, 1998) and size exclusion chromatography (Kotaria et al, 1999) occurred because the contributions of the bound detergent and lipid had not been properly accounted for (Bamber et al, 2006; Crichton et al, 2013; Kunji et al, 2008). Here, we also show that none of the studied carriers are cross-linked when they are expressed in the inner membrane of mitochondria and isolated in a folded form (Figs. 1A,B and 2). In agreement, isolated bovine OGC is also not cross-linked, although cross-linking can be induced artificially with $Cu^{2+}$-phenanthroline or diamide (Bisaccia et al, 1996). Finally, in systematic mutagenesis studies, no conserved dimerization interface was detected, accordant with OGC being a structural monomer (Cappello et al, 2006; Cappello et al, 2007; Miniero et al, 2011), as also shown experimentally here (Fig. 2). Thus, apart from AGC, which has an additional dimerization domain, all mitochondrial carriers are likely to be monomeric in agreement with their structural features (Bamber et al, 2006; Bamber et al, 2007b; Crichton et al, 2013; Harborne et al, 2015; Kunji et al, 2008; Lee et al, 2015).

All mitochondrial carriers have the same functional elements that are important for the transport mechanism, such as the matrix salt bridge network (Pebay-Peyroula et al, 2003; Robinson et al, 2008; Ruprecht et al, 2014), glutamine braces (Ruprecht et al, 2014), the cytoplasmic salt bridge network (King et al, 2016; Robinson et al, 2008; Ruprecht et al, 2014; Ruprecht et al, 2019), tyrosine braces (Ruprecht et al, 2019), small residues in the interhelical interfaces (Kunji et al, 2020; Ruprecht et al, 2019; Ruprecht and Kunji, 2020) and a single central substrate-binding site (Kunji and Robinson, 2006; Mavridou et al, 2022; Robinson and Kunji, 2006; Robinson et al, 2008). Thus, all mitochondrial carriers have the required properties to function as monomers as well, as shown experimentally for the mitochondrial ADP/ATP carrier (Bamber et al, 2007a). In the context of the mitochondrial inner membrane, which has a high protein density, these carriers exist as monomers (except for AGC2) and function as monomers, forming only weak, nonspecific, and transient protein interactions that are of no consequence.

Here, we revisited the kinetic mechanism of four diverse human carriers by using a straightforward experimental system in which the purified carriers are reconstituted into proteoliposomes, allowing full control of the chemical gradients of the substrates. Uptake curves were determined by robotics, meaning that all recorded data points follow the same procedure. We show that the human mitochondrial oxoglutarate, citrate, dicarboxylate, and aspartate/glutamate carriers all operate with a ping-pong kinetic mechanism (Fig. 3), as previously shown for the mitochondrial ADP/ATP carrier (Cimadamore-Werthein et al, 2023). A particularly interesting case is the mitochondrial aspartate/glutamate carrier, which is a structural dimer (Fig. 2) (Thangaratnarajah et al, 2014), but nonetheless functions with a ping-pong kinetic mechanism (Fig. 3K,L). The explanation is that the carrier domains do not interact, as observed before (Thangaratnarajah et al, 2014), allowing them to function independently of each other.

It is difficult to explain the fundamental differences between current and previously published kinetic analyses, but there might be some contributing factors. In some studies, whole mitochondria were used (Sluse et al, 1973), which are not ideal as the substrate gradients are difficult to control and other transport proteins with overlapping substrate specificities might be present, for example, for malate, sulfate, or glutamate. In early kinetic studies, mitochondrial carriers were often purified by negative chromatography, such as hydroxyapatite columns with Triton X-100 or X-114 (Bisaccia et al, 1993; Bisaccia et al, 1990; Indiveri et al, 1991; Indiveri et al, 1993). Under these conditions, the carriers pass through the column, because the lipid/detergent micelle completely shields it from the column materials, preventing it from binding (Kunji et al, 2008). The problem is that neither the conditions (e.g., lipid content) nor the purity (e.g. the presence of other mitochondrial carriers of similar size) can be fully controlled, meaning other transport activities cannot be excluded. Another important experimental issue can be the removal of the external radiolabeled substrate to determine the amount of accumulated radiolabel inside the proteoliposomes. In most of the procedures, gel filtration columns were used, such as Sephadex G-75, which separate very slowly. Thus, to prevent the release of radiolabel, specific or unspecific inhibitors or cocktails thereof were used to quench the transport, adding a differential competitive factor to the kinetic analysis. In many cases, the procedures to determine the initial rates are not transparently described, but often the "zero" time point follows a different procedure from the "transport" time point. Some aspects of data gathering and analysis were also insufficient,

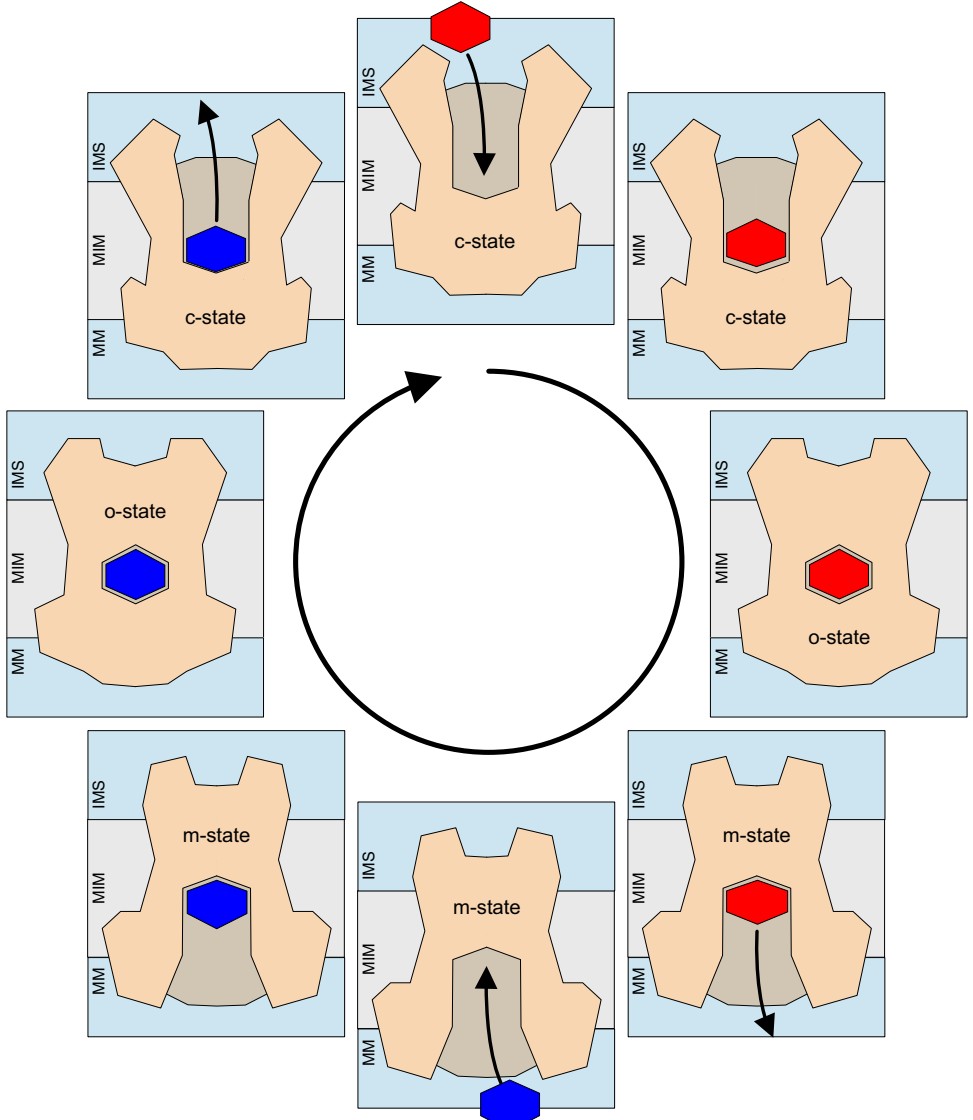

**Figure 4. Schematic representation of the ping-pong mechanism of mitochondrial carrier proteins.**

Conformational changes between the cytoplasmic-open state (c-state), occluded state (o-state), and matrix-open state (m-state) with the carriers shown schematically, viewed laterally from the membrane. The imported substrate is shown in red and the exported substrate in blue. MM mitochondrial matrix, MIM mitochondrial inner membrane, IMS intermembrane space.

as often only three internal concentrations were used (Bisaccia et al, 1993; Indiveri et al, 1991), no error analysis was carried out (Bisaccia et al, 1993; Dierks et al, 1988; Indiveri et al, 1991; Indiveri et al, 1993) and in nearly all cases the kinetic assessment was overly reliant on double reciprocal (Lineweaver–Burk) plots. These plots have the fundamental flaw in that the slope strongly depends on the less accurately determined points obtained at low concentrations, whereas the more accurately determined points at high concentrations cluster. Furthermore, in some cases, the chosen buffers may have interfered with the kinetic analyses. For example, the ethane sulphonic-acid PIPES buffer was used as a component in transport assays with the dicarboxylate carrier (Indiveri et al, 1989a; Indiveri et al, 1989b; Indiveri et al, 1993), which had previously been shown to transport sulfate (Crompton et al, 1974) and could therefore

provide a differentially competing factor. In our procedures, all of these potential pitfalls were avoided.

The demonstration that mitochondrial carriers operate with ping-pong kinetics implies that one substrate dissociates from the carrier before the counter-substrate binds for transport in the opposite direction (double-displacement). It is consistent with an alternating-access mechanism, in which a transport protein opens a single substrate-binding site alternately to one or the other side of the membrane via an occluded state (Fig. 4). This type of mechanism was first proposed for the bacterial phosphate translocase by Mitchell, as early as 1957 (Mitchell, 1957), although Jardetzky is often credited, proposing a similar mechanism for the P-type Na⁺-K⁺ pump, published in 1966 (Jardetzky, 1966). However, it was the analysis of the effect of specific inhibitors on

substrate transport by the mitochondrial ADP/ATP carrier that provided the first experimental proof for an alternating-access mechanism, renamed by Klingenberg as the "single binding center gated pore mechanism" (Klingenberg, 1976). Atractyloside and carboxyatractyloside (Luciani et al, 1971; Vignais et al, 1971) lock the carrier in the cytoplasmic-open state (Erdelt et al, 1972; Klingenberg and Buchholz, 1973), in which the substrate-binding site is open to the mitochondrial intermembrane space, whereas bongkrekic acid locks the carrier in the matrix-open state, in which the substrate-binding site is open to the mitochondrial matrix (Erdelt et al, 1972; Klingenberg and Buchholz, 1973). The structures of the inhibited cytoplasmic-open state (Kunji and Harding, 2003; Pebay-Peyroula et al, 2003; Ruprecht et al, 2014) and matrix-open state (Ruprecht et al, 2019) provided direct structural support for this notion. In the case of the mitochondrial ADP/ATP carrier, the structural mechanism is only consistent with a ping-pong kinetic mechanism (Cimadamore-Werthein et al, 2023), based on a monomer with a single substrate-binding site (Bamber et al, 2006; Bamber et al, 2007a; Bamber et al, 2007b; Crichton et al, 2013; Kunji et al, 2008; Mavridou et al, 2022). Here, we show that the human mitochondrial oxoglutarate, citrate, dicarboxylate and aspartate/glutamate carriers also operate by ping-pong reaction kinetics (Fig. 3) and thus function with an alternating-access mechanism.

Despite having different structures and mechanisms, most, if not all, transport proteins have an alternating-access mechanism (Drew and Boudker, 2016). The only outstanding claim for a sequential mechanism which we are aware of is for a member of the hetero-dimeric L-amino acid transporter family (SLC7) (Reig et al, 2007). However, structural analysis suggests that these transport proteins have a single central substrate-binding site that is alternatingly accessible (Errasti-Murugarren et al, 2019), indicative of a ping-pong kinetic mechanism. It is highly probable that all members of the mitochondrial carrier family have an alternating-access transport mechanism, based on them being functional monomers, operating with ping-pong kinetics (Cimadamore-Werthein et al, 2023; Kunji et al, 2020; Kunji and Ruprecht, 2020; Ruprecht et al, 2014; Ruprecht et al, 2019; Ruprecht and Kunji, 2020, 2021).

# Methods

## Expression of human mitochondrial carriers in *Saccharomyces cerevisiae*

Genes encoding the human oxoglutarate carrier (OGC, SLC25A11, UniProt accession code: Q02978), the human citrate carrier (CIC, SLC25A1, UniProt accession code: P53007), the human dicarboxylate carrier (DIC, SLC25A10, UniProt accession code: Q9UBX3) and the human aspartate/glutamate carrier 2 (AGC2, SLC25A13, UniProt accession code: Q9UJS0) were codon-optimized with an N-terminal eight histidine tag and either Xa (OGC, CIC and DIC) or TEV (AGC2) protease cleavage sites to aid purification (OGC, Δ2-13; CIC, Δ2-12; DIC, GKP added (positions 2–4)) (synthesized by GenScript). These constructs were cloned into the pYES2/CT vector (Invitrogen), and transformed into yeast strain W303.1B (OGC, CIC and AGC2) or the protease-deficient yeast strain BJ2168 (DIC) (Gietz and Schiestl, 2007; King and Kunji, 2020).

Successful transformants were selected on Sc-Ura + 2% (w/v) glucose plates. For large-scale expression, a pre-culture of cells grown in Sc-Ura + 2% (w/v) glucose was inoculated into 50 L of YPG + 0.1% glucose medium in an Applikon Pilot Plant 140-L bioreactor. Cells were grown at 30 °C for 16 h, induced with 0.4% galactose, grown for either 4 h (OGC and AGC2) or 8 h (CIC and DIC), and harvested by centrifugation (4000 × g, 20 min, 4 °C). Crude mitochondria were prepared using a bead mill (Dyno-Mill Multilab, Willy A. Bachofen AG) by established methods (King and Kunji, 2020). The human uncoupling protein (UCP1) and ADP/ATP carrier (AAC1) were expressed and purified, as described previously (Cimadamore-Werthein et al, 2023; Jaiquel Baron et al, 2021; Jones et al, 2023).

## Preparation of lipids for protein purification

Tetraoleoyl cardiolipin (18:1) in powder-form (Avanti Polar Lipids), was dissolved in a 10% (w/v) solution of lauryl maltose neopentyl glycol (Anatrace) by vortexing for 4 h, resulting in a concentration of 10 mg mL$^{-1}$ lipid in a 10% (w/v) detergent solution. These prepared stocks were stored in liquid nitrogen.

## Purification of human mitochondrial carriers by nickel affinity chromatography

Yeast mitochondria expressing CIC were first washed in alkali buffer (100 mM Na$_2$CO$_3$, pH 11.5, 1 mM EDTA) for 30 min to remove peripherally-associated proteins, centrifuged (25,000 × g, 1 h, 4 °C), resuspended in Tris-buffered glycerol (100 mM Tris-HCl, pH 7.4, 10% glycerol), centrifuged again and resuspended in Tris-buffered glycerol. Aliquots were snap-frozen in liquid nitrogen.

Crude mitochondria were solubilized in a solution containing 1.5% (w/v) lauryl maltose neopentyl glycol, EDTA-free protease inhibitor tablets (Roche), and 150 mM NaCl. This was complemented by varying concentrations of imidazole: 30 mM for AGC2, 40 mM for OGC and DIC, or 50 mM for CIC. The mixture was then mixed by rotation at 4 °C, for 1 h (OGC, DIC, and CIC) or 2 h (AGC2). The soluble fraction was separated from the insoluble material by centrifugation (200,000 × g, 45 min, 4 °C). Nickel sepharose slurry (GE Healthcare), at a volume of 0.7 mL per 1 g crude mitochondria, was then added to the supernatant. This mixture was stirred at 4 °C for 1 h (OGC, DIC, and CIC) or 2 h (AGC2). The nickel resin was collected by centrifugation (100 × g, 10 min, 4 °C) and transferred to a Proteus 1-step batch midi spin column (Generon). The resin was washed by centrifugation (100 × g, 15 min, 4 °C): 30 column volumes of buffer A (20 mM HEPES pH 7.0, 150 mM NaCl, 40 mM imidazole, 0.1 mg mL$^{-1}$ tetraoleoyl cardiolipin/0.1% (w/v) lauryl maltose neopentyl glycol), followed by 10 column volumes of buffer B (20 mM HEPES pH 7.0, 50 mM NaCl, 0.1 mg mL$^{-1}$ tetraoleoyl cardiolipin/0.1% (w/v) lauryl maltose neopentyl glycol).

For OGC and DIC, the nickel resin was resuspended with 0.7 mL of buffer B and incubated with 20 mM imidazole, 30 μg factor Xa protease (NEB), and 5 mM CaCl$_2$. The on-column cleavage was performed overnight at 4 °C with rotation. The proteins were then collected by centrifugation (500 × g, 2 min, 4 °C), and the samples were desalted using a midi PD10 desalting column (GE Healthcare), pre-equilibrated with buffer B. For AGC2,

the nickel resin was resuspended in an equal volume of buffer B (0.7 mL) and incubated with 60 mM imidazole, 25 µg TEV (Tobacco Etch Virus protease), and 1.5 mM DTT overnight at 10 °C with rotation. The protein was eluted by centrifugation (500 × $g$, 2 min, 4 °C), and imidazole was removed using 2 mL Zeba Spin Desalting Columns (Thermo Fisher Scientific). TEV was subsequently removed by incubation with amylose resin for 20 min in the cold room, and the resin was removed by centrifugation as before. For CIC, the nickel resin was dispensed into a poly-prep column (Bio-Rad), allowed to settle under gravity, and protein was eluted with 2.5 mL buffer (20 mM HEPES pH 7.0, 150 mM NaCl, 400 mM imidazole, 0.2 mg mL$^{-1}$ tetraoleoyl cardiolipin/0.2% (w/v) lauryl maltose neopentyl glycol). Imidazole was removed immediately by desalting with a PD10 column.

The protein concentration in the elution fractions was measured by spectrometry (NanoDrop Technologies) at 280 nm (OGC: extinction coefficient; 29,000 M$^{-1}$ cm$^{-1}$, protein mass; 33,477 Da. CIC: extinction coefficient; 31,400 M$^{-1}$ cm$^{-1}$, protein mass; 32,713 Da. DIC: extinction coefficient; 20,900 M$^{-1}$ cm$^{-1}$, protein mass; 31,923 Da. AGC2, other protein (E 1%) = 6.04). The purity and stability of the final samples were assessed by gel electrophoresis and thermostability shift assays.

## Thermostability analysis

Differential scanning fluorimetry (nanoDSF) was used to assess the thermostability of proteins (Alexander et al, 2014). In this study, the proteins were characterized by their tryptophan content: OGC (two tryptophans); CIC (three tryptophans); DIC (one tryptophan); AGC2 (three tryptophans). For the analysis, 10 µg of protein was mixed into 10 µL of buffer B, with or without 10 mM of substrate. NanoDSF-grade standard glass capillaries were employed to load the samples into the Prometheus NT.48 nanoDSF instrument. The temperature setting was progressively increased from 25 °C to 95 °C, at a rate of 4 °C per minute. The software PR.ThermControl (NanoTemper Technologies) was used to determine the proteins' apparent melting temperatures (Tm).

## Size exclusion chromatography

All carriers were analysed by size exclusion chromatography (SEC). In brief, 150 µL of purified AAC1 (0.7 mg/mL), AGC2 (1.9 mg/mL), CIC (1.2 mg/mL), DIC (1.7 mg/mL), OGC (1.1 mg/mL), and UCP1 (1.6 mg/mL) were applied to a Superdex 200 Increase 10/300 SEC column (GE Healthcare) equilibrated with purification buffer B (except for UCP1: Jones et al, 2023) on an ÄKTA explorer system. The determination of the apparent molecular weight of each carrier was done using a gel filtration calibration kit (GE Healthcare), as indicated in the legend of Fig. 2. The structural models in Fig. 2A were generated by the Alphafold 3.0 server (Abramson et al, 2024), except for the experimentally determined UCP1 structure (Jones et al, 2023).

## Reconstitution of protein into liposomes

A reconstitution unit included 12.6 mg of total lipids (9 mg from *E. coli* polar lipid extract, 3 mg egg L-α-phosphatidylcholine and 0.6 mg tetraoleoyl cardiolipin, from Avanti Polar Lipids) with

30–75 µg of protein, in a 1.25 mL final volume. Each reconstituted sample was used for transport experiments involving one internal and eight external concentrations. Lipids were hydrated in a cold solution of 50 mM NaCl and 20 mM HEPES (pH 7.0 for OGC and CIC), 20 mM Tris (pH 7.0 for DIC), or 20 mM HEPES (pH 7.4 for AGC2), and solubilized with pentaethylene glycol monodecyl ether ($C_{10}E_5$). $C_{10}E_5$ was removed by five sequential additions (4x 60 mg and 1x 480 mg) of SM-2 bio-beads (Bio-Rad) over 20 min. The sample was incubated overnight at 4 °C under rotation. The bio-beads were then separated using micro-bio spin columns (Bio-Rad). Substrates (malate for OGC and DIC, citrate for CIC, or aspartate for AGC2) at varying concentrations (0, 0.10, 0.25, 0.50, and 1.00 mM), along with 5 mM MgCl$_2$ (for OGC, CIC, and DIC) or 1 mM CaCl$_2$ (for AGC2), were internalized through three freeze-thaw cycles and proteoliposome formation by extrusion (21 passes through a 0.45-µm filter). External substrates were removed via PD10 desalting columns (GE Healthcare) previously equilibrated with transport buffer (20 mM HEPES pH 7.0, 50 mM NaCl, 5 mM MgCl$_2$ for OGC and CIC, 20 mM HEPES pH 7.4, 50 mM NaCl, 1 mM CaCl$_2$ for AGC2, or 20 mM Tris pH 7.0, 50 mM NaCl, 5 mM MgCl$_2$ for DIC). Finally, 1.5 mL desalted proteoliposomes were diluted six times in buffer for transport assays.

## Transport assays

Transport experiments were performed using the Hamilton MicroLab Star robot (Hamilton Robotics Ltd). Diluted proteoliposomes (100 µL) were pipetted into the wells of MultiScreenHTS + HA 96-well filter plates (pore size 0.45-µm, Millipore). For the human oxoglutarate carrier, uptake of radiolabeled [$^{14}$C]-malate (American Radiolabeled Chemicals) was initiated by the addition of 100 µL buffer containing 0.5, 1.0, 1.5, 2.5, 5.0, 10, 15, or 20 µM [$^{14}$C]-malate. For the human citrate carrier, uptake of radiolabeled [$^{14}$C]-citrate was initiated by the addition of 100 µL buffer containing 1.0, 1.5, 2.5, 5.0, 10, 15, 20, or 25 µM [$^{14}$C]-citrate (Perkin Elmer). For the human dicarboxylate carrier, uptake of radiolabeled [$^{14}$C]-malate was initiated by the addition of 100 µL buffer containing 1.0, 1.5, 2.5, 5.0, 10, 15, 20, 25, or 50 µM [$^{14}$C]-malate. For the human aspartate/glutamate carrier, uptake of radiolabeled [$^{14}$C]-aspartate (American Radiolabeled Chemicals) was initiated by the addition of 100 µL buffer containing 1.5, 2.5, 5.0, 10, 15, 20, or 25 µM [$^{14}$C]-aspartate. Uptake was stopped at 0, 10, 20, 30, 40, 50, 60, 90, and 120 s by filtration and washing with 200 µL transport buffer three times. Plates were dried overnight, after which 200 µL MicroScint-20 (Perkin Elmer) were added, and radioactivity levels were determined using a TopCount scintillation counter (Perkin Elmer). Apparent $K_m$ and $V_{max}$ values were determined by fitting Michaelis–Menten curves in Prism.

## Protein quantification

Protein standards and proteoliposome samples were run on SDS-polyacrylamide gel electrophoresis at 180 V using 4–12% gradient gels (mPAGE, Merck). The gels were treated with a 40% ethanol and 10% acetic acid fixing solution at room temperature for 2 h, then stained overnight at 4 °C with Flamingo fluorescent gel stain (Bio-Rad). Gel imaging was performed on an Amersham Typhoon (GE Healthcare) with a Cy2 emission filter, using settings of

500–550 PMT (V) and a pixel size of 25 μm. ImageQuant TL software (Toolbox v8.1) analyzed the images, utilizing the rolling ball algorithm for background subtraction and band volumes to estimate protein quantities in samples against a standard curve of 10 to 100 ng of protein.

## Data availability

This study includes no data deposited in external repositories.

The source data of this paper are collected in the following database record: biostudies:S-SCDT-10_1038-S44318-024-00150-0.

## Peer review information

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

## Acknowledgements

This work was supported by the Medical Research Council, grant MC_UU_00028/2 to MSK and ERSK. CC-W was supported by an Aker Scholarship, DL by the Swiss National Science Foundation (Synergia project CRSII5_180326), EP by the Norges Forskningsråd (Forskningsrådet)(grant 301170), and SJB by a BBSRC-GSK CASE Fellowship. We thank Dr. Shane Palmer for multiple large-scale fermentation runs of yeast, required for this project.

## Author contributions

**Camila Cimadamore-Werthein**: Conceptualization; Data curation; Formal analysis; Investigation; Visualization; Methodology; Writing—original draft; Writing—review and editing. **Martin S King**: Conceptualization; Data curation; Formal analysis; Validation; Investigation; Visualization; Methodology; Writing—original draft; Writing—review and editing. **Denis Lacabanne**: Conceptualization; Data curation; Formal analysis; Validation; Investigation; Visualization; Methodology; Writing—original draft; Writing—review and editing. **Eva Pyrihová**: Data curation; Investigation; Methodology; Writing—review and editing. **Stephany Jaiquel Baron**: Data curation; Investigation; Methodology; Writing—review and editing. **Edmund RS Kunji**: Conceptualization; Resources; Data curation; Formal analysis; Supervision; Funding acquisition; Validation; Investigation; Visualization; Methodology; Writing—original draft; Project administration; Writing—review and editing.

Source data underlying figure panels in this paper may have individual authorship assigned. Where available, figure panel/source data authorship is listed in the following database record: biostudies:S-SCDT-10_1038-S44318-024-00150-0.

## Disclosure and competing interests statement

The authors declare no competing interests.

# Expanded View Figures

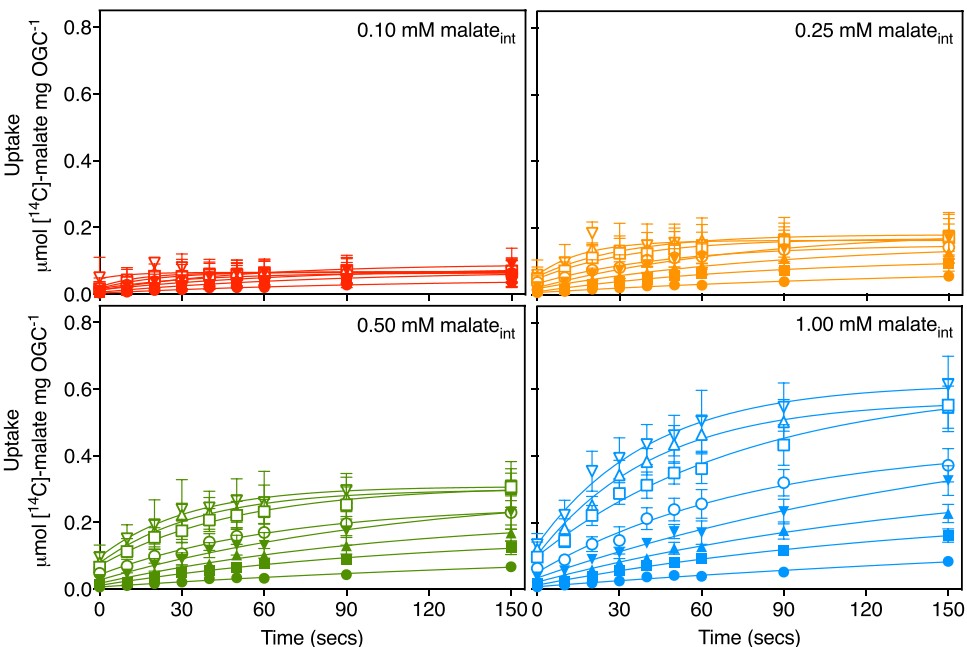

**Figure EV1.   Uptake of transport catalyzed by reconstituted human oxoglutarate carrier.**

Proteoliposomes containing human oxoglutarate carrier were loaded with either 0.10 mM (red traces), 0.25 mM (orange traces), 0.50 mM (green traces), or 1.00 mM (blue traces) malate$_{int}$, and the transport was initiated by the externally added radiolabeled malate at either 0.5 μM (filled circles), 1.0 μM (filled squares), 1.5 μM (filled upward triangles), 2.5 μM (filled downward triangles), 5.0 μM (open circles), 10 μM (open squares), 15 μM (open upward triangles), or 20 μM (open downward triangles) [$^{14}$C]-malate (malate$_{ext}$). Initial rates were estimated by fitting the uptake data to Eq. 1. The data represent the average and standard deviation of $n = 6$ (two independent experiments, each with three technical repeats). Source data are available online for this figure.

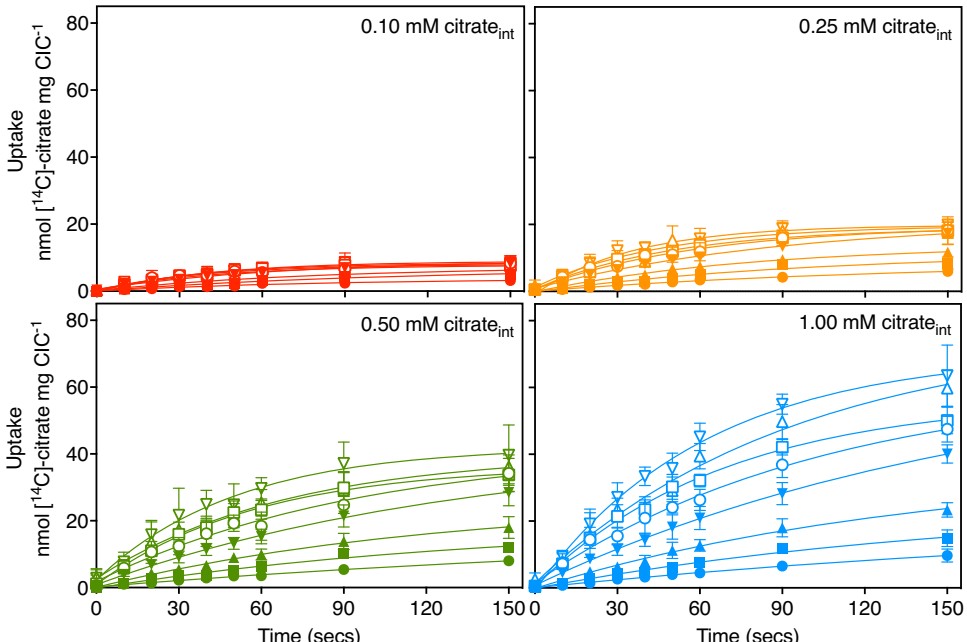

**Figure EV2. Uptake curves of the transport catalyzed by reconstituted human citrate carrier.**

Proteoliposomes containing human citrate carrier were loaded with either 0.10 mM (red traces), 0.25 mM (orange traces), 0.50 mM (green traces), or 1.00 mM (blue traces) citrate$_{int}$, and transport was initiated by the externally added radiolabeled sulfate at either 1.0 µM (filled circles), 1.5 µM (filled squares), 2.5 µM (filled upward triangles), 5 µM (filled downward triangles), 10 µM (open circles), 15 µM (open squares), 20 µM (open upward triangles), or 25 µM (open downward triangles) [$^{14}$C]-citrate (citrate$_{ext}$). Initial rates were estimated by fitting the uptake data to Eq. 1. The data represent the average and standard deviation of $n = 6$ (two independent experiments, each with three technical repeats). Source data are available online for this figure.

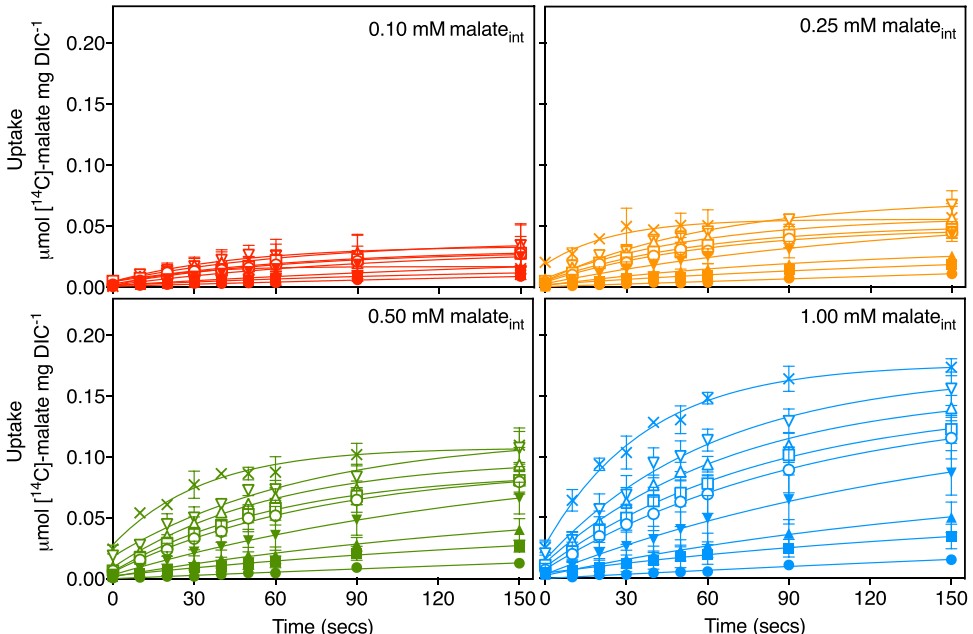

**Figure EV3.   Uptake curves of the transport catalyzed by reconstituted human dicarboxylate carrier.**

Proteoliposomes containing human dicarboxylate carrier were loaded with either 0.10 mM (red traces), 0.25 mM (orange traces), 0.50 mM (green traces), or 1.00 mM (blue traces) malate$_{int}$, and transport was initiated by the externally added radiolabeled malate at either 1.0 µM (filled circles), 1.5 µM (filled squares), 2.5 µM (filled upward triangles), 5 µM (filled downward triangles), 10 µM (open circles), 15 µM (open squares), 20 µM (open upward triangles), 25 µM (open downward triangles), or 50 µM (crosses) [$^{14}$C]-malate (malate$_{ext}$). Initial rates were estimated by fitting the uptake data to Eq. 1. The data represent the average and standard deviation of $n = 6$ (two independent experiments, each with three technical repeats, except the 1 and 50 µM external [$^{14}$C]-malate datasets, which are the average of three technical repeats). Source data are available online for this figure.

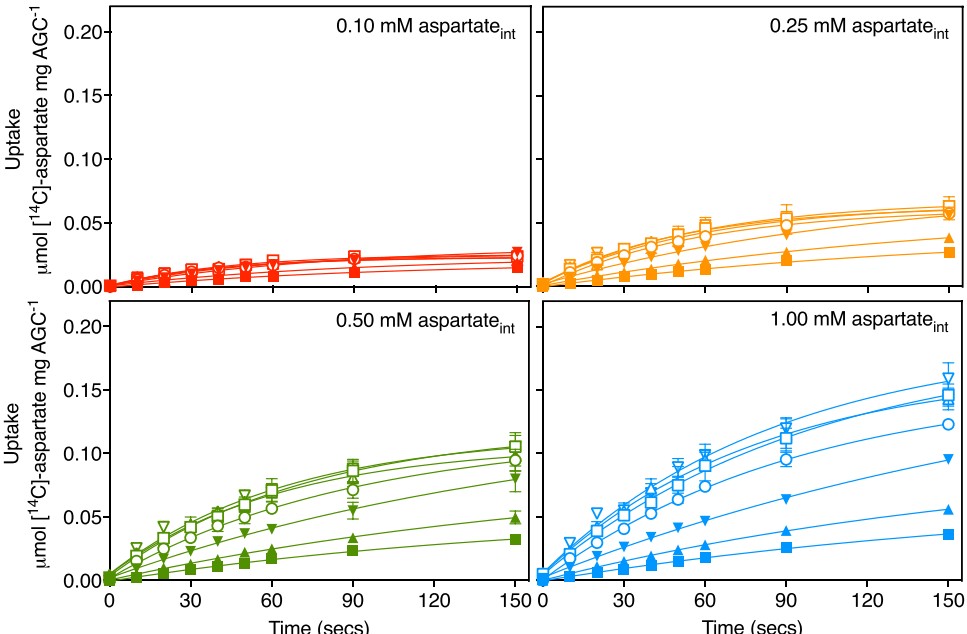

**Figure EV4. Uptake curves of the transport catalyzed by reconstituted human aspartate/glutamate carrier.**

Proteoliposomes containing human aspartate/glutamate carrier were loaded with either 0.10 mM (red traces), 0.25 mM (orange traces), 0.50 mM (green traces), or 1.00 mM (blue traces) aspartate$_{int}$, and the transport was initiated by the externally added radiolabeled aspartate at either 1.5 µM (filled squares), 2.5 µM (filled upward triangles), 5.0 µM (filled downward triangles), 10 µM (open circles), 15 µM (open squares), 20 µM (open upward triangles), or 25 µM (open downward triangles) [$^{14}$C]-aspartate (aspartate$_{ext}$). Initial rates were estimated by fitting the uptake data to Eq. 1. The data represent the average and standard deviation of three technical repeats. Source data are available online for this figure.

