## [Peer Review File · The EMBO Journal]

Most human mitochondrial carriers of the SLC25 family function as monomers exchanging substrates with a ping-pong kinetic mechanism

Camila Cimadamore-Werthein, Martin King, Denis Lacabanne, Eva Pyrihová, Stephany Jaiquel Baron, and Edmund Kunji

Corresponding author(s): Edmund Kunji (ek@mrc-mbu.cam.ac.uk)

Review Timeline:

Submission Date:	14th Mar 24
Editorial Decision:	17th Apr 24
Revision Received:	26th Apr 24
Editorial Decision:	27th May 24
Revision Received:	30th May 24
Accepted:	31st May 24

Editor: Ioannis Papaioannou

Transaction Report:

Dear Edmund,

Thank you again for submitting your manuscript EMBOJ-2024-117289 for consideration by The EMBO Journal. It has been seen by three experts in the field, and we have received the full set of their comments, which I have already shared with you (they are included again below). I would also like to thank you for taking the time to discuss with me your results and the referees' comments, which was very helpful for us to reach a fair and balanced editorial decision on the manuscript.

The referees recognize that the quality of the data is excellent, and the results are solid and clearly presented in the manuscript. They also have a few suggestions for clarification or further improvement of the manuscript, and they encourage you to discuss your results in the context of the published literature regarding the relevant question of the ping-pong vs. sequential kinetic mechanism of the analyzed carriers as well as the debate on their monomeric vs. dimeric/oligomeric nature.

Given the referees' comments and recommendations, as well as our conversation, I would like to invite you to submit a revised version of the manuscript along with a detailed point-by-point response addressing all referees' comments. I should add that it is EMBO Journal policy to allow only a single round of major revision, and acceptance of your manuscript will therefore depend on the completeness of your responses in this revised version. Please let me know if you have any questions or comments.

We generally allow three months as standard revision time (July 16, 2024). As a matter of policy, competing manuscripts published during this period will not negatively impact our assessment of the conceptual advance presented by your study. However, we request that you contact us as soon as possible upon publication of any related work, to discuss how to proceed. Should you foresee a problem in meeting this three-month deadline, please let us know in advance and we may be able to grant an extension.

Thank you for the opportunity to consider your work for publication in The EMBO Journal. I look forward to your revision.

Best regards,

Ioannis

Instructions for preparing your revised manuscript

1. When you are ready to submit the revision, please upload:

- A Word file of the manuscript text (including legends of main Figures, EV Figures and Tables). Please make sure that changes are highlighted (or "tracked") to be clearly visible.

- Individual production-quality figure files (one file per figure). When assembling your figures, please refer to our figure preparation guidelines in order to ensure proper formatting and readability in print as well as on screen:

If the data shown in a figure are obtained from n {less than or equal to} 2, please use scatter plots showing the individual data points.

- i. the name of the statistical test used to generate error bars and P values
- ii. the number (n) of independent experiments (please specify technical or biological replicates) underlying each data point (discussion of statistical methodology can be reported in the Materials and Methods section, but figure legends should contain a basic description of n , P , and the test applied)
- iii. the nature of the bars and error bars (s.d., s.e.m.).

- A point-by-point response to the referees' comments, with a detailed description of the changes made (as a word file). All referees' concerns must be fully addressed and their suggestions taken on board. When preparing your letter of response to the

referees' comments, please bear in mind that this will form part of the Review Process File and will therefore be available online to the community. Please note that you have the possibility to opt out of the transparent process at any stage prior to publication by letting the editorial office know (contact@embojournal.org); if you do opt out, the Review Process File link will point to the following statement: "No Review Process File is available with this article, as the authors have chosen not to make the review process public in this case.". For more details on our Transparent Editorial Process, please visit our website: <https://www.embopress.org/page/journal/14602075/authorguide#transparentprocess>

- Expanded View (EV) files (replacing Supplementary Information) that are collapsible/expandable online. A maximum of 5 EV Figures can be typeset. EV Figures should be cited as "Figure EV1, Figure EV2" etc. in the text, and their respective legends should be included in the manuscript file after the legends of regular figures. See detailed instructions regarding Expanded View files here:

- For the figures that you do NOT wish to display as Expanded View figures, they should be bundled together with their legends in a single PDF file called "Appendix", which should start with a short Table of Contents (including page numbers). Appendix figures should be referred to in the main text as: "Appendix Figure S1, Appendix Figure S2" etc. Please see detailed instructions here: <https://www.embopress.org/page/journal/14602075/authorguide#expandedview>

- A complete author checklist, which you can download from our author guidelines (<https://www.embopress.org/page/journal/14602075/authorguide>). Please note that the checklist will also be part of the Review Process File.

2. Please note that no statistics should be calculated and shown in Figures if $n=2$.

3. Before submitting your revision, primary datasets (and computer code, where appropriate) produced in this study need to be deposited in appropriate public databases (see <https://www.embopress.org/page/journal/14602075/authorguide#dataavailability>). The accession numbers and databases should be listed in a formal "Data availability" section (placed after Materials and Methods) that follows the model below (see also <https://www.embopress.org/page/journal/14602075/authorguide#dataavailability>):

Data availability

- RNA-seq data: Gene Expression Omnibus GSE46843 (<https://www.ncbi.nlm.nih.gov/geo/query/acc.cgi?acc=GSE46843>)
- [data type]: [name of the resource] [accession number/identifier/doi] ([URL or identifiers.org/DATABASE:ACCESSION])

*** All links should resolve to a page where the data can be accessed. ***

*** Please remember to provide in the Data availability section of your revised manuscript reviewer passwords if the datasets are not yet public. ***

*** The Data Availability Section is restricted to new primary data that are part of this study. In case you have no data that require deposition in a public database, please state so instead of referring to the database: "Our study includes no data deposited in public repositories." under the heading "Data availability". ***

4. Please check that the title and the abstract of the manuscript are brief, yet explicit, even to non-specialists. The length of the title should not exceed 100 characters, and the abstract should be a single paragraph not exceeding 175 words.

5. Please also note our reference format: <https://www.embopress.org/page/journal/14602075/authorguide#referencesformat>.

7. Please remember: digital image enhancement is acceptable practice, as long as it accurately represents the original data and conforms to community standards. If a figure has been subjected to significant electronic manipulation, this must be noted in the figure legend or in the "Materials and Methods" section. The editors reserve the right to request original versions of figures and the original images that were used to assemble the figure.

8. Our journal encourages inclusion of data citations in the reference list to directly cite datasets that were obtained from public databases. Data citations in the article text are distinct from normal bibliographical citations and should directly link to the database records from which the data can be accessed. In the main text, data citations are formatted as follows: "Data ref: Smith et al, 2001" or "Data ref: NCBI Sequence Read Archive PRJNA342805, 2017". In the Reference list, data citations must

be labeled with "[DATASET]". A data reference must provide the database name, accession number/identifiers, and a resolvable link to the landing page from which the data can be accessed at the end of the reference. Further instructions are available at: <https://www.embopress.org/page/journal/14602075/authorguide#referencesformat>.

9. We request authors to consider both actual and perceived competing interests. Please review our policy (<https://www.embopress.org/page/journal/14602075/authorguide#conflictsofinterest>) and update your competing interests statement if necessary. Please name this section 'Disclosure and competing interests statement' and place it after the Acknowledgements section.

10. Please note that all corresponding authors are required to provide an ORCID ID upon submission of a revised manuscript (<https://orcid.org/>). Please find instructions on how to link your ORCID ID to your account in our manuscript tracking system in our Author guidelines (<https://www.embopress.org/page/journal/14602075/authorguide#authorshipguidelines>).

11. We use CRediT to specify the contributions of each author in the journal submission system. CRediT replaces the author contribution section, which should be removed from the manuscript. Please use the free text box to provide more detailed descriptions. See also guide to authors: <https://www.embopress.org/page/journal/14602075/authorguide#authorshipguidelines>.

13. We would also welcome the submission of cover suggestions or motifs to be used by our Graphics Illustrator in designing a cover.

14. Please use the link below to submit your revision:
<https://emboj.msubmit.net/cgi-bin/main.plex>

Referee #1:

The manuscript by Cimadamore et al expands upon their earlier work on the ATP/ADP carrier (Cimadamore-Werthein et al., 2023) to assay the oligomeric state and transport kinetics of human mitochondrial SLC25 transporter family members A1, A10, A11, and A13 (citrate, dicarboxylate, oxoglutarate, and aspartate/glutamate carrier, respectively). The authors first perform thermal stability tests on their purified transporters expressed in yeast mitochondrial membranes with and without ligands and confirm the expected ligand mediated stabilization of their proteins that reflects normal physiological binding. The transporters are then subjected to the same robotic liposome-based substrate transport assay developed to study SLC25A4 and confirm that all four transporters follow a ping-pong/double displacement transport mechanism where K_m/V_{max} ratios are independent of counter substrate concentrations as opposed to a sequential mechanism that has also been invoked for this class of transporters but one that is largely inconsistent with structural data. The authors go on to assay the oligomeric state of the transporters using SEC to show that with the exception of SLC27A13, the only one of the transporters investigated here to harbor a dimerization domain, function as monomers. The experimental work is excellent, the manuscript clearly presented, and the results further support the ping-pong mechanism based on functional monomers. This is consequential given the importance of these transporters and ongoing debate about their mechanistic features underlying substrate transport. The following points could be considered to further strengthen and clarify the manuscript for publication:

1. It is unclear to what extent there is still an ongoing debate about the ping-pong vs sequential mechanisms, with the former supported by a large body of structural and functional studies, including the Kunji group's earlier work. As written, one gets the sense the authors are relitigating largely settled questions that, considering the physiological importance of these proteins, detracts from the novelty of the current results in moving the field forward by working out their detailed mechanisms of substrate transport.
2. The SEC data highlight differences between A13 and the rest of the transporters under investigation but are inherently limited considering retention times are reliant on hydrodynamic radius. Higher oligomeric states may therefore have overlapping elution profiles with a dimer. If definitive statements on the oligomeric state are to be made, the conclusions could be better served by addition of more quantitative analysis such as native MS or AUC.
3. Inclusion of a simple figure with structural alignments of the 4 transporters (experimental evidence and/or predicted structures) as supplementary/extended data would be beneficial to non SLC25 family specialists to highlight dimerization domains and other functionally relevant aspects.
4. Relatedly, where does oligomerization fit into the mechanistic framework depicted in figure 3?

Referee #2:

The manuscript by Cimadamore-Werthein et al. constitutes a thorough and elegant functional characterization of some members of the SLC25 mitochondrial carrier family. Specifically, the authors investigated a fundamental question concerning the mode of action, i.e., whether the tested SLC25 members transport their respective substrate according to a sequential kinetic model or a so-called ping-pong (or double-displacement) kinetic mechanism.

The work is part of the continuous effort by the Kunji group to uncover the molecular mechanisms underlying essential transport processes across the mitochondrial inner membrane. The experimental design of this study follows a recent publication (EMBO Reports, PMID: 37278158) by the same group that focused on the kinetic characterization of the ADP/ATP carrier SLC25A4. Although the experimental approach is not new - in fact, 'classic' transport measurements and analyses were applied here - the execution of the experiments (a tour de force) is quite impressive. The results are solid, thoroughly analyzed, and clearly presented. The manuscript demonstrates an excellent level of quality, it is written in a professional and clear language, and the results are well-illustrated and presented in a conclusive manner. The key findings of the work - concluding that the tested members of the SLC25 family operate according to a ping-pong mechanism - are interesting and are well supported by the data. Overall, the manuscript is of high quality, and I would recommend this article for publication, after a minor revision.

Specific comments:

The authors focus their analysis on the mitochondrial carriers OGC, CIC, DIC, and AGC2 and provide detailed information on the substrate specificity for each of these transporters on pages 4 & 5 of the manuscript, pointing out that they exchange different substrates (e.g., AGC2 imports Glu in exchange (export) with Asp). However, the experiments shown in Fig 1 were conducted with only the same substrate inside and outside of the proteoliposomes (in the case for AGC2, non-labeled Asp inside, ¹⁴C-Asp on the outside). It is important to clarify for non-experts in the field that these carriers also exchange the same substrate to better understand the design of the uptake experiments depicted in Fig. 1.

Minor comment:

Line 170: include a space between 'ping-pongor'

Referee #3:

In this manuscript, the authors have provided enough data to support a ping-pong mechanism of some mitochondrial carriers. However, the reviewer is concerned that the authors did not focus only on the kinetic property, rather than mixing it too much with the oligomeric state of the carriers. Although the ping-pong mechanism through the central cavity of the carriers indicates the possibility of a functional monomer, it does not mean that the carriers can only exist as monomer. On the other hand, existing in a dimeric form does not mean that the dimer has to function following the previously proposed sequential mechanism which the authors are against. In fact, in addition to a huge body of early evidence, more recent in situ mass spectrometry (Chorev et al, 2018) also supports the dimeric form of the ADP/ATP carrier. The reviewer doesn't understand why the authors insist so strongly on the monomer proposal. When so much evidence is not compatible with the proposed model, possibly we should spend more time thinking about adjusting the model rather than doubt on the previously published results. Considering the extreme crowdedness of the inner mitochondrial membrane that facilitates aggregation and the tightly bound negative cardiolipins around the mitochondrial carriers that facilitates separation, it seems reasonable to think that dynamic equilibrium exists between the monomer and the dimeric form, and such equilibrium should be very sensitive to the change in the lipid composition during protein purification and reconstitution. That may partly explain why different groups observed different oligomeric states of these carriers.

In brief, as the oligomeric condition is still an open question of most intensive debate and controversy in this field, the reviewer highly recommends the authors to focus only on the kinetic problem in this manuscript.

Referee #1:

The manuscript by Cimadamore et al expands upon their earlier work on the ATP/ADP carrier (Cimadamore-Werthein et al., 2023) to assay the oligomeric state and transport kinetics of human mitochondrial SLC25 transporter family members A1, A10, A11, and A13 (citrate, dicarboxylate, oxoglutarate, and aspartate/glutamate carrier, respectively). The authors first perform thermal stability tests on their purified transporters expressed in yeast mitochondrial membranes with and without ligands and confirm the expected ligand mediated stabilization of their proteins that reflects normal physiological binding. The transporters are then subjected to the same robotic liposome-based substrate transport assay developed to study SLC25A4 and confirm that all four transporters follow a ping-pong/double displacement transport mechanism where K_m/V_{max} ratios are independent of counter substrate concentrations as opposed to a sequential mechanism that has also been invoked for this class of transporters but one that is largely inconsistent with structural data. The authors go on to assay the oligomeric state of the transporters using SEC to show that with the exception of SLC27A13, the only one of the transporters investigated here to harbor a dimerization domain, function as monomers. The experimental work is excellent, the manuscript clearly presented, and the results further support the ping-pong mechanism based on functional monomers. This is consequential given the importance of these transporters and ongoing debate about their mechanistic features underlying substrate transport.

We thank the reviewer for their positive comments on our manuscript.

The following points could be considered to further strengthen and clarify the manuscript for publication:

1. It is unclear to what extent there is still an ongoing debate about the ping-pong vs sequential mechanisms, with the former supported by a large body of structural and functional studies, including the Kunji group's earlier work. As written, one gets the sense the authors are relitigating largely settled questions that, considering the physiological importance of these proteins, detracts from the novelty of the current results in moving the field forward by working out their detailed mechanisms of substrate transport.

We would have hoped that the extensive structural and functional data in the literature would have put the 'monomer-dimer' and 'ping-pong-sequential' arguments to bed, but unfortunately this is not the case. Many in the field still believe that the carriers are functional dimers (as highlighted by the comments reviewer 3), and that they have a sequential kinetic mechanism. Only this month a published paper claims that the mitochondrial oxoglutarate carrier (SLC25A11) is dimeric in SDS gels (Zuna *et al*, 2024), but the protein was isolated from inclusion bodies produced in *E. coli*. Here we show that mitochondrial oxoglutarate carrier is a monomer, when isolated from the inner membrane, and is not cross-linked, just like the bovine carrier (Bisaccia *et al*, 1996).

The majority of the extant literature claims that mitochondrial carriers function with a sequential kinetic mechanism. This study, along with the equivalent analysis of the mitochondrial ADP/ATP carrier (Cimadamore-Werthein *et al*, 2023), is the first unifying

investigation of the kinetic mechanism of this family of proteins, which is consistent with an alternating access mechanism in agreement with the structural data.

2. The SEC data highlight differences between A13 and the rest of the transporters under investigation but are inherently limited considering retention times are reliant on hydrodynamic radius. Higher oligomeric states may therefore have overlapping elution profiles with a dimer. If definitive statements on the oligomeric state are to be made, the conclusions could be better served by addition of more quantitative analysis such as native MS or AUC.

We thank the reviewer for their comments. The provided SEC data are conclusive with regards to the oligomeric state of the carriers, as there are no alternative explanations than the one provided.

First of all, the elution volumes of OGC, DIC, CIC, are similar to those of AAC and UCP1 for which accurate size determinations have been carried using a range of biophysical techniques (Bamber *et al*, 2006; Bamber *et al*, 2007; Lee *et al*, 2015), which are also in agreement with direct structural observations, showing that they are monomers (Jones *et al*, 2023; Ruprecht *et al*, 2019b).

Second, we have shown with a large number of detergents that the weight contributions of lipid, detergent and protein are simply additive to the total weight in SEC (Bamber *et al*, 2006; Kunji *et al*, 2008). Using a series of markers, which do not bind detergent or lipid, we have determined that the total weights of the lipid/detergent/protein complex of OGC, DIC, CIC, AAC, UCP1 are slightly larger than aldolase (158 kDa), i.e. ~165 kDa, whereas the total weight of AGC2 is slightly larger than ferritin (440 kDa), i.e. ~450 kDa. They were all purified under the same conditions in laurylmaltoseneopentylglycol (LMNG) plus tetraoleoylcardiolipin (TOCL). The total weights can be explained well by a ~135 kDa detergent/lipid contribution, which is similar that of a pure dodecylmaltoside contribution (115 kDa) (Bamber *et al*, 2006; Kunji *et al*, 2008), but here TOCL was also added, which will enlarge the micelle further (Kunji & Crichton, 2010). The ~135-kDa lipid/detergent contribution would mean that the determined weights of the carriers are about $165 - 135 = 30$ kDa, which matches well with their molecular weights (see figure. The determined weight for AGC2 is about $450 - 2 * 135 = 180$ kDa for two protomers or 90 kDa for each, which again compares well with its molecular weight of 74 kDa.

Third, had they been dimers then the total bound lipid/detergent contribution would have to have been $165 - 2 * 33 = 99$ kDa or ~50 kDa for each 'protomer', which is even smaller than a pure bound octylmaltoside micelle, which is 67 kDa (Bamber *et al*, 2006; Kunji *et al*, 2008), and thus an impossibility.

Fourth, the peak for any purely hypothetical dimer ($2 * 33 + 2 * 135 = 336$ kDa), now indicated with a dotted line, would have been much closer to that of the AGC2 (450 kDa) than the observed peaks of the monomers.

Thus, it is a physical impossibility that OGC, DIC and CIC are dimers in SEC, once the contributions of bound lipid/detergent to the total weights are properly accounted for. We have provided more details of the rationale.

3. Inclusion of a simple figure with structural alignments of the 4 transporters (experimental evidence and/or predicted structures) as supplementary/extended data would be beneficial to non SLC25 family specialists to highlight dimerization domains and other functionally relevant aspects.

We agree with the reviewer and have included the structures as part of the SEC data figure, now moved into the main part of the manuscript. A comparison shows the obvious relationship between the structures and the SEC data. Only AGC2 has a dimerization domain and indeed runs as a dimer.

Figure 2: Structures and oligomeric states of mitochondrial carriers. (A) Structural models of the human dicarboxylate carrier (DIC), oxoglutarate carrier (OGC), citrate carrier (CIC), ADP/ATP carrier (AAC1), and aspartate/glutamate carrier (AGC2) were all determined by the AlphaFold 3.0 server (Abramson et al., 2024), except for the experimentally determined structure of uncoupling protein (UCP1) (pdb entry: 8G8W) (Jones et al., 2023). (B) Determination of the molecular weights in LMNG/TOCL by size-exclusion chromatography. The normalized absorbance traces for monomeric AAC1 (blue), DIC (red), OGC (green), CIC (cyan), UCP1 (brown) and dimeric AGC2 (purple) are shown. The standards used for sizing are ferritin (440 kDa), aldolase (158 kDa), conalbumin (76 kDa), and ovalbumin (43 kDa). The dotted line indicates the elution volume of a hypothetical dimer peak for DIC, OGC and CIC.

4. Relatedly, where does oligomerization fit into the mechanistic framework depicted in figure 3?

The mechanism depicted in figure 3 shows a carrier functioning as a monomer, which is consistent with the mechanistic as well as the kinetic and sizing data, presented in this paper. In the context of the high-density inner membrane, the carrier would simply exist and function as a monomer forming only weak, non-specific and transient interactions with other proteins that are of no consequence. We have clarified this point in the text.

Referee #2:

The manuscript by Cimadamore-Werthein et al. constitutes a thorough and elegant functional characterization of some members of the SLC25 mitochondrial carrier family. Specifically, the authors investigated a fundamental question concerning the mode of action, i.e., whether the tested SLC25 members transport their respective substrate according to a sequential kinetic model or a so-called ping-pong (or double-displacement) kinetic mechanism. The work is part of the continuous effort by the Kunji group to uncover the molecular mechanisms underlying essential transport processes across the mitochondrial inner membrane. The experimental design of this study follows a recent publication (EMBO Reports, PMID: 37278158) by the same group that focused on the kinetic characterization of the ADP/ATP carrier SLC25A4. Although the experimental approach is not new - in fact, 'classic' transport measurements and analyses were applied here - the execution of the experiments (a tour de force) is quite impressive. The results are solid, thoroughly analyzed, and clearly presented. The manuscript demonstrates an excellent level of quality, it is written in a professional and clear language, and the results are well-illustrated and presented in a conclusive manner. The key findings of the work - concluding that the tested members of the SLC25 family operate according to a ping-pong mechanism - are interesting and are well supported by the data. Overall, the manuscript is of high quality, and I would recommend this article for publication, after a minor revision.

We thank the reviewer for their positive comments on our manuscript.

Specific comments:

The authors focus their analysis on the mitochondrial carriers OGC, CIC, DIC, and AGC2 and provide detailed information on the substrate specificity for each of these transporters on pages 4 & 5 of the manuscript, pointing out that they exchange different substrates (e.g., AGC2 imports Glu in exchange (export) with Asp). However, the experiments shown in Fig 1 were conducted with only the same substrate inside and outside of the proteoliposomes (in the case for AGC2, non-labeled Asp inside, 14C-Asp on the outside). It is important to clarify for non-experts in the field that these carriers also exchange the same substrate to better understand the design of the uptake experiments depicted in Fig. 1.

We thank the reviewer for this suggestion. Mitochondrial carriers are fully reversible transporters and can catalyse 'hetero-exchange' and, as used here, 'homo-exchange'

reactions in which the exchanged substrates are chemically the same, but the external one is radiolabelled. We have added a sentence in the manuscript to explain this.

Minor comment:

Line 170: include a space between 'ping-pong'

We thank the reviewer for noting this typographical error.

Referee #3:

In this manuscript, the authors have provided enough data to support a ping-pong mechanism of some mitochondrial carriers. However, the reviewer is concerned that the authors did not focus only on the kinetic property, rather than mixing it too much with the oligomeric state of the carriers. Although the ping-pong mechanism through the central cavity of the carriers indicates the possibility of a functional monomer, it does not mean that the carriers can only exist as monomer. On the other hand, existing in a dimeric form does not mean that the dimer has to function following the previously proposed sequential mechanism which the authors are against. In fact, in addition to a huge body of early evidence, more recent *in situ* mass spectrometry (Chorev *et al*, 2018) also supports the dimeric form of the ADP/ATP carrier. The reviewer doesn't understand why the authors insist so strongly on the monomer proposal. When so much evidence is not compatible with the proposed model, possibly we should spend more time thinking about adjusting the model rather than doubt on the previously published results. Considering the extreme crowdedness of the inner mitochondrial membrane that facilitates aggregation and the tightly bound negative cardiolipins around the mitochondrial carriers that facilitates separation, it seems reasonable to think that dynamic equilibrium exists between the monomer and the dimeric form, and such equilibrium should be very sensitive to the change in the lipid composition during protein purification and reconstitution. That may partly explain why different groups observed different oligomeric states of these carriers.

In brief, as the oligomeric condition is still an open question of most intensive debate and controversy in this field, the reviewer highly recommends the authors to focus only on the kinetic problem in this manuscript.

We note that the reviewer is insisting that we should remove crucial information on the oligomeric state of mitochondrial carriers to prevent us from making well-supported contributions to the 'intensive' debate. The reason for including the SEC data is that there are a lot of incorrect claims in the literature with regards to the dimeric state of mitochondrial carriers, unnecessarily confusing the situation. Here, we show that OGC, CIC, and DIC are all monomers in contrast to earlier claims, which were due to analytical errors. For example, it was not realised that carriers run with bound detergent/lipid in native electrophoresis and SEC, and thus their contributions were not properly accounted for (Crichton *et al*, 2013; Kunji & Crichton, 2010). The real issue is that there never have been valid claims of a dimer (except for AGC2). We have moved the SEC data as a figure in the main part of the text, showing also the basic structures of these proteins, as suggested by reviewer 2.

We have already refuted the incorrect dimer claim of Chorev *et al.*, here (Hirst *et al.*, 2019) and in more detail here (Kunji & Ruprecht, 2020). It is a simple case of mistaken identity, as the native bovine ADP/ATP carrier does not have a mass 33,195 Da, as claimed by Chorev *et al.*, but a mass of 32,921 Da, as determined independently by amino acid sequencing (Aquila *et al.*, 1982; Babel *et al.*, 1981; Klingenberg, 1989) and mass spectrometry (Smith *et al.*, 2003), agreeing also with the known structure of the same protein (Pebay-Peyroula *et al.*, 2003). There are much better candidates for the 33,195 Da mass in mitochondria had they bothered to check. In a new unreviewed manuscript by the same authors, available on BioRxiv (<https://doi.org/10.1101/2023.05.05.539595>), native mass spectrometry is used to confirm that yeast Aac2p is a monomer with bound cardiolipins, as we had already correctly demonstrated many, many, many years ago with the same methods as used in this manuscript (Bamber *et al.*, 2006; Ruprecht *et al.*, 2014a).

It has been known for many decades that the mitochondrial inner membrane has a high protein density. Thus, we do not doubt that carriers will come into contact with each other from time to time. The key questions are whether these interactions are always the same, as strictly required for a dimer definition, and whether they are functionally relevant. The carriers have all of the structural and mechanistic elements to function as monomers (Kunji & Ruprecht, 2020; Ruprecht & Kunji, 2019; Ruprecht & Kunji, 2020, 2021) and they are structural monomers (Jones *et al.*, 2023; Kang & Chen, 2023; Pebay-Peyroula *et al.*, 2003; Ruprecht *et al.*, 2014b; Ruprecht *et al.*, 2019a). The only confirmed exception is AGC2 which has a unique N-terminal dimerization domain and is an actual structural dimer (Thangaratnarajah *et al.*, 2014). In the context of the high-density inner membrane, the carriers would simply exist and function as monomers (but for AGC2), forming only weak, non-specific and transient interactions with other proteins that are of no consequence.

Aquila H, Misra D, Eulitz M, Klingenberg M (1982) Complete amino acid sequence of the ADP/ATP carrier from beef heart mitochondria. *Hoppe Seylers Z Physiol Chem* 363: 345-349

Babel W, Wachter E, Aquila H, Klingenberg M (1981) Amino acid sequence determination of the ADP/ATP carrier from beef heart mitochondria. The sequence of the C-terminal acidolytic fragment. *Biochim Biophys Acta* 670: 176-180

Bamber L, Harding M, Butler PJG, Kunji ERS (2006) Yeast mitochondrial ADP/ATP carriers are monomeric in detergents. *Proc Natl Acad Sci USA* 103: 16224-16229

Bamber L, Slotboom DJ, Kunji ERS (2007) Yeast mitochondrial ADP/ATP carriers are monomeric in detergents as demonstrated by differential affinity purification. *J Mol Biol* 371: 388-395

Bisaccia F, Zara V, Capobianco L, Iacobazzi V, Mazzeo M, Palmieri F (1996) The formation of a disulfide cross-link between the two subunits demonstrates the dimeric structure of the mitochondrial oxoglutarate carrier. *Biochim Biophys Acta* 1292: 281-288

Cimadamore-Werthein C, Jaiquel Baron S, King MS, Springett R, Kunji ER (2023) Human mitochondrial ADP/ATP carrier SLC25A4 operates with a ping-pong kinetic mechanism. *EMBO Rep*: e57127

Crichton PG, Harding M, Ruprecht JJ, Lee Y, Kunji ERS (2013) Lipid, detergent, and Coomassie Blue G-250 affect the migration of small membrane proteins in blue native gels; mitochondrial carriers migrate as monomers not dimers. *J Biol Chem* 288: 22163-22173

Hirst J, Kunji ERS, Walker JE (2019) Comment on "Protein assemblies ejected directly from native membranes yield complexes for mass spectrometry". *Science* 366: 700-700

Jones SA, Gogoi P, Ruprecht JJ, King MS, Lee Y, Zögg T, Pardon E, Chand D, Steimle S, Copeman DM *et al* (2023) Structural basis of purine nucleotide inhibition of human uncoupling protein 1. *Sci Adv* 9: eadh4251

Kang Y, Chen L (2023) Structural basis for the binding of DNP and purine nucleotides onto UCP1. *Nature* 620: 226-231

Klingenberg M (1989) Molecular aspects of the adenine nucleotide carrier from mitochondria. *Arch Biochem Biophys* 270: 1-14

Kunji ERS, Crichton PG (2010) Mitochondrial carriers function as monomers. *Biochim Biophys Acta* 1797: 817-831

Kunji ERS, Harding M, Butler PJG, Akamine P (2008) Determination of the molecular mass and dimensions of membrane proteins by size exclusion chromatography. *Methods* 46: 62-72

Kunji ERS, Ruprecht JJ (2020) The mitochondrial ADP/ATP carrier exists and functions as a monomer. *Biochem Soc Trans* 48: 1419-1432

Lee Y, Willers C, Kunji ERS, Crichton PG (2015) Uncoupling protein 1 binds one nucleotide per monomer and is stabilized by tightly bound cardiolipin. *Proc Natl Acad Sci U S A* 112: 6973-6978

Pebay-Peyroula E, Dahout-Gonzalez C, Kahn R, Trezeguet V, Lauquin GJ, Brandolin G (2003) Structure of mitochondrial ADP/ATP carrier in complex with carboxyatractyloside. *Nature* 426: 39-44

Ruprecht JJ, Hellawell AM, Harding M, Crichton PG, McCoy AJ, Kunji ER (2014a) Structures of yeast mitochondrial ADP/ATP carriers support a domain-based alternating-access transport mechanism. *Proc Natl Acad Sci U S A* 111: E426-434

Ruprecht JJ, Hellawell AM, Harding M, Crichton PG, McCoy AJ, Kunji ERS (2014b) Structures of yeast mitochondrial ADP/ATP carriers support a domain-based alternating-access transport mechanism. *Proc Natl Acad Sci U S A* 111: E426-E434

Ruprecht JJ, King MS, Zogg T, Aleksandrova AA, Pardon E, Crichton PG, Steyaert J, Kunji ERS (2019a) The molecular mechanism of transport by the mitochondrial ADP/ATP carrier. *Cell* 176: 435-447

Ruprecht JJ, King MS, Zögg T, Aleksandrova AA, Pardon E, Crichton PG, Steyaert J, Kunji ERS (2019b) The molecular mechanism of transport by the mitochondrial ADP/ATP carrier. *Cell* 176: 435-447

Ruprecht JJ, Kunji ER (2019) Structural changes in the transport cycle of the mitochondrial ADP/ATP carrier. *Curr Opin Struct Biol* 57: 135-144

Ruprecht JJ, Kunji ERS (2020) The SLC25 mitochondrial carrier family: structure and mechanism. *Trends Biochem Sci* 45: 244-258

Ruprecht JJ, Kunji ERS (2021) Structural mechanism of transport of mitochondrial carriers *Annu Rev Biochem* 90: 535-558

Smith VR, Fearnley IM, Walker JE (2003) Altered chromatographic behaviour of mitochondrial ADP/ATP translocase induced by stabilization of the protein by binding of 6'-O-fluorescein-atractyloside. *Biochem J* 376: 757-763

Thangaratnarajah C, Ruprecht JJ, Kunji ERS (2014) Calcium-induced conformational changes of the regulatory domain of human mitochondrial aspartate/glutamate carriers. *Nat Commun* 5: 5491

Zuna K, Tyschuk T, Beikbaghban T, Sternberg F, Kreiter J, Pohl EE (2024) The 2-oxoglutarate/malate carrier extends the family of mitochondrial carriers capable of fatty acid and 2,4-dinitrophenol-activated proton transport. *Acta Physiol*

Dear Edmund,

Thank you for the submission of your revised manuscript to The EMBO Journal and your patience. We have now received the comments of the three referees that were asked to re-assess your study (included below). As you will see, all referees are satisfied with the revision and now support publication of the manuscript in our journal.

I would also like to thank you for your offer to update the manuscript with the results of the recently released new version of the AlphaFold program. Could you please update the manuscript and the figure accordingly?

There are also a few minor changes and corrections that we need from you before we can proceed with acceptance of the manuscript:

- You can now please remove "tracked" changes from the Word file of your manuscript.
- Please enter all relevant funding information in our online manuscript handling system. This information should be identical to that provided in the Acknowledgements section of your manuscript (currently missing from the online system: Aker Scholarship; and missing from the manuscript: UKRI | Biotechnology and Biological Sciences Research Council (BBSRC) CASE Studentship).
- Please note that only up to 5 keywords can be listed (you currently have 7).
- Please change the heading of your conflict-of-interest statement to "Disclosure and competing interests statement".
- We noticed that the "Appendix Figure S1" -which does not exist- is called out in your manuscript. Please correct the callout.
- Thank you for providing source data for your Figures. Our source data coordinator will send you shortly further instructions related to the requested source data and how to organize them.
- Please note that EMBO press papers are accompanied online by:
 - A) a short (2 sentences) summary of the findings and their significance,
 - B) 2-5 short bullet points highlighting the key results, and
 - C) a synopsis image in .jpg or .png format that is exactly 550 pixels wide and 300-600 pixels high (the height is variable). Please note that the text needs to be legible at the final size. Please upload this information along with your revised manuscript (the text for A and B should be provided in a separate Word file).

Please also note that as part of the EMBO publications' Transparent Editorial Process, The EMBO Journal publishes online a Peer Review File along with each accepted manuscript. This File will be published in conjunction with your paper and will include the referee reports, your point-by-point response and all pertinent correspondence relating to the manuscript. You can opt out of this by letting the editorial office know (contact@embojournal.org). If you do opt out, the Peer Review File link will point to the following statement: "No Peer Review File is available with this article, as the authors have chosen not to make the review process public in this case."

We look forward to seeing a final version of your manuscript as soon as possible. Please use this link to submit your revision:
<https://emboj.msubmit.net/cgi-bin/main.plex>

Best regards,

Ioannis

Referee #1:

The authors have addressed the minor concerns I had raised.

Referee #2:

Given that I had only minor revision requests, the authors addressed these in satisfactory manner. I suggest acceptance of the revised manuscript for publication.

Referee #3:

The reviewer don't mind the current version being published in the journal, although the reviewer keeps the opinion that the oligomeric state is still far from being settled unless more convincing and direct evidence such as those from AFM appear.

All editorial and formatting issues were resolved by the authors.

Dear Edmund,

Congratulations on an excellent manuscript! I am very pleased to inform you that it has been accepted for publication in The EMBO Journal. Thank you for your comprehensive responses to the referee concerns.

If you have any questions, please do not hesitate to contact the Editorial Office. Thank you for your contribution to The EMBO Journal. It has been a pleasure working with you.

Best wishes,

Ioannis
